# LMCleaner: Efficient and Certified Online Unlearning via Influence Propagation Truncation

Jie Xu [1]   Zihan Wu [1]   Wenbo Pan [1]   Jiao Yin [2]   Yong-Feng Ge [2]   Hua Wang [2]   Cong Wang [1]   Xiaohua Jia [1]

## Abstract

Existing machine unlearning methods primarily focus on removing data influence after training completes, which is effective for many scenarios, but a complementary capability is needed when removal requests arise during ongoing training. We propose LMCleaner, an efficient and certified *online* unlearning framework that can process unlearning requests at any training step without waiting for training completion. Our key insight is that influence propagation can be decomposed into a trust region where linear approximation is accurate, and a residual that concentrates in a low-dimensional subspace and can be efficiently masked by calibrated noise. Building on this insight, we design an influence propagation truncation mechanism that treats mini-batch influence as atomic units, computes influence within a truncation window for efficient removal, and injects subspace-aware noise for certified privacy. Our theoretical analysis proves that the truncation residual decays exponentially with window size and that the unlearned model is $(\varepsilon, \delta)$-indistinguishable from retraining. Experiments demonstrate that LMCleaner achieves over $100\times$ computational savings compared to baselines while maintaining model utility and defending against membership inference attacks.

## 1. Introduction

Large Language Models (LLMs) are trained on web-scale datasets. Despite extensive pre-processing, malicious data, such as poisoned samples, backdoor triggers, or harmful content, may enter the training pipeline (Nguyen et al., 2025).

Recent advances in data auditing and anomaly detection enable identifying such problematic data *during* training (Xu et al., 2025a), but by the time detection occurs, the malicious influence has already begun propagating through model parameters.

Most machine unlearning works focus on removing data influence *after* training completion, using techniques such as influence-function-based modifications (Guo et al., 2020), negative fine-tuning (Zhang et al., 2024), and output filtering (Liu et al., 2024). These post-training approaches have achieved remarkable success for their target scenarios. However, when malicious data is detected mid-training, waiting for convergence allows its influence to propagate through all subsequent parameter updates, compounding the contamination. This motivates an *online* unlearning framework that can remove data influence at any training step, immediately upon detection, without waiting for model convergence.

It is challenging to achieve online unlearning for LLMs. 1) The storage overhead required for influence removal is prohibitive. Training data affects the model by contributing gradients to parameter updates. To remove this influence, gradients of all data points must be recorded, resulting in memory costs that scale linearly with both model size and dataset size. 2) The computational cost of influence propagation is impractical. The influence of training data propagates through all subsequent parameter updates. Exactly computing this propagated influence requires tracing through all parameter updates, which is intractable when training spans millions of steps. 3) There is no theoretical mechanism for LLMs certified unlearning. Existing mechanisms rely on convex optimization theory or exact Hessian inversion (Guo et al., 2020), both of which are inapplicable to the non-convex, billion-parameter landscape of LLMs.

To address these challenges, we propose LMCleaner, an efficient and certified online unlearning framework that can process unlearning requests at any training step without waiting for convergence. For the storage challenge, we adopt a mini-batch unlearning strategy that treats mini-batches as atomic units, since all data points within a mini-batch jointly induce a single parameter update, and their influences are inherently entangled. This strategy reduces storage cost by over 500× compared to the sample-level

[1]Department of Computer Science, City University of Hong Kong, Hong Kong SAR, China [2]Institute for Sustainable Industries and Liveable Cities, Victoria University, Melbourne, Australia. Correspondence to: Jie Xu <jiexu49-c@my.cityu.edu.hk>.

*Proceedings of the $43^{rd}$ International Conference on Machine Learning*, Seoul, South Korea. PMLR 306, 2026. Copyright 2026 by the author(s).

method. For the computational challenge, we design an influence propagation truncation mechanism that computes influences only within a small truncation window (e.g., 50 steps). This mechanism significantly reduces computational overhead, avoiding millions of computation steps. For certified privacy protection, we inject calibrated Gaussian noise to efficiently mask residual influence, achieving certified $(\varepsilon, \delta)$-unlearning indistinguishable from retraining.

Overall, the main contributions are summarized as follows:

- We propose LMCLEANER, the first online unlearning framework practically scalable to LLM training, removing data influence *during training* without waiting for model convergence.

- We design an influence propagation truncation mechanism that treats mini-batches as atomic units, computes influence within a truncation window, and injects calibrated noise for $(\varepsilon, \delta)$-certified unlearning.

- Experiments demonstrate over $100\times$ computational savings compared to baselines while maintaining model utility and defending against Membership Inference Attacks.

**Conflict of Interest Disclosure.** The authors declare no financial conflicts of interest.

## 2. Related Work

Machine unlearning methods can be broadly classified into exact unlearning and approximate unlearning (Xu et al., 2024). Exact unlearning aims to achieve a model state identical to one trained from scratch on the remaining dataset. The representative approach is SISA (Bourtoule et al., 2021), which divides data into shards, trains independent sub-models, and selectively retrains only affected sub-models. Various derivatives have emerged: DC-k-means (Ginart et al., 2019), DaRE (Brophy & Lowd, 2021) and HedgeCut (Schelter et al., 2021) for tree-based models, GraphEraser (Chen et al., 2022b) and RecEraser (Chen et al., 2022a) for graph models, and LMEraser (Xu et al., 2025b) for large models. While providing complete data removal guarantees, exact methods scale prohibitively with model size and complexity (Xu et al., 2024).

Approximate unlearning improves efficiency by reducing rather than exactly removing unlearned data influence. Early approaches used influence functions (Guo et al., 2020; Sekhari et al., 2021) to quantify and reverse data impact, though non-convexity in optimization challenges their practical application. GA (Wu et al., 2020; Neel et al., 2021) and WGA (Wang et al., 2025) are negative fine-tuning approaches that directly increase the loss on the forget set via gradient ascent. However, their limited precision, particularly with complex or overlapping data, often degrades the general utility of the model. Negative Preference

Optimization (NPO) approaches (Zhang et al., 2024; Fan et al., 2025) treat forgotten data as negative preferences, adjusting parameters to assign low likelihoods to the forget set while maintaining proximity to the original model. They typically rely on a retain set to preserve model utility, which conflicts with data protection regulations (Voigt & Von dem Bussche, 2024). Model editing techniques (Jang et al., 2023; Wu et al., 2023; Jin et al., 2024) target specific knowledge by applying localized parameter modifications or activation interventions. Output filtering methods (Liu et al., 2024; Bhaila et al., 2025) operate at inference time without parameter changes, though they may leave internal representations unchanged. These post-training approaches have significantly advanced the field and remain the method of choice when unlearning is performed after training completion. Our work explores a complementary direction: online unlearning during training, which may offer advantages in scenarios requiring immediate response to deletion requests before model convergence. While HFR also supports certified online deletion, its need to maintain and update per-sample influence vectors throughout training makes it impractical for billion-parameter LLMs. In contrast, LMCLEANER computes influence on demand within a short truncation window, making online certified unlearning practical at LLM scale.

## 3. Preliminaries and Problem Definition

### 3.1. Preliminaries

Let $\mathcal{Z} = \mathcal{X} \times \mathcal{Y}$ denote the data space, where $\mathcal{X} \subseteq \mathbb{R}^d$ is the input domain, $\mathcal{Y}$ is the output space, and the training dataset is $D = \{z_i = (x_i, y_i)\}_{i=1}^N$. We assume single-pass training where each $z_i$ is used exactly once. The extension to multi-epoch fine-tuning is developed in Appendix D.

At step $t$, the model parameters are $\theta^{[t]} \in \mathbb{R}^p$ and the sampled mini-batch is $S_t \subseteq \{1, \ldots, N\}$ with $|S_t| = B$. We denote $f_{\theta^{[t]}}$ as the model function at step $t$. For fair comparison with prior unlearning methods (Bourtoule et al., 2021; Ginart et al., 2019), the model is trained using mini-batch Stochastic Gradient Descent (SGD) where $\ell(z; \theta)$ is the loss and $\eta_t$ is the learning rate:

$$\theta^{[t+1]} = \theta^{[t]} - \frac{\eta_t}{|S_t|} \sum_{i \in S_t} \nabla_\theta \ell(z_i; \theta^{[t]}). \tag{1}$$

For convenience, we define the mini-batch gradient as $\bar{g}^{[t]} := \frac{1}{|S_t|} \sum_{i \in S_t} \nabla_\theta \ell(z_i; \theta^{[t]})$ and the mini-batch Hessian as $H^{[t]} := \frac{1}{|S_t|} \sum_{i \in S_t} \nabla_\theta^2 \ell(z_i; \theta^{[t]})$.

### 3.2. Certified Unlearning

**Definition 3.1** $((\varepsilon, \delta)$-Certified Unlearning (Guo et al., 2020)). An unlearning algorithm $\mathcal{A}$ achieves $(\varepsilon, \delta)$-certified unlearning if for any dataset $D$, any training data point

$z \in D$, and any measurable set $W \subseteq \mathbb{R}^p$:

$$\Pr[\mathcal{A}(D, z) \in W] \leq e^{\varepsilon} \Pr[\mathcal{R}(D \setminus \{z\}) \in W] + \delta, \quad (2)$$

$$\Pr[\mathcal{R}(D \setminus \{z\}) \in W] \leq e^{\varepsilon} \Pr[\mathcal{A}(D, z) \in W] + \delta, \quad (3)$$

where $\mathcal{R}(D')$ denotes retraining from scratch on dataset $D'$. This ensures the unlearned model is statistically indistinguishable from retraining within privacy budget $(\varepsilon, \delta)$.

### 3.3. Problem Definition

We aim to design an *online* unlearning method that processes unlearning requests immediately during training, achieving certified privacy guarantees while preserving model utility.

Formally, at any training step $\tau$, an unlearning request may arrive for target data $D_u \subset D$ that appeared in earlier steps. The *ideal unlearned model* $f_{\text{ideal}}$ is defined as a model retrained from scratch on the retained data $D' = D \setminus D_u$ following the same training protocol (learning rate, batch size, etc.). The unlearning method aims to obtain an unlearned model $f'$ that behaves as $f_{\text{ideal}}$, preserves utility on $D'$, satisfies $(\varepsilon, \delta)$-certified unlearning, and enables efficient execution without full retraining.

## 4. Design of LMCLEANER

We design LMCLEANER based on 1) sample-level influence propagation as the theoretical foundation (Section 4.1), 2) mini-batch unlearning for storage efficiency (Section 4.2), and 3) truncated propagation with noise injection for efficient and certified unlearning (Section 4.3). Figure 1 provides an overview of the full pipeline.

### 4.1. Sample-Level Influence Propagation

Consider a training data point $z_j$ used at step $t_{z_j}$, now requested for unlearning at step $\tau \geq t_{z_j} + 1$. The standard parameter update follows:

$$\theta^{[t+1]} = \theta^{[t]} - \eta_t \bar{g}^{[t]}, \qquad \bar{g}^{[t]} = \frac{1}{|S_t|} \sum_{i \in S_t} \nabla_\theta \ell(z_i; \theta^{[t]}). \quad (4)$$

The counterfactual update excluding $z_j$[1] is $\theta_{-j}^{[t+1]} = \theta_{-j}^{[t]} - \eta_t \bar{g}_{-j}^{[t]}$, where

$$\bar{g}_{-j}^{[t]} := \begin{cases} \dfrac{1}{|S_{t_{z_j}}|} \displaystyle\sum_{i \in S_{t_{z_j}} \setminus \{j\}} \nabla_\theta \ell\left(z_i; \theta_{-j}^{[t]}\right), & t = t_{z_j}, \\ \dfrac{1}{|S_t|} \displaystyle\sum_{i \in S_t} \nabla_\theta \ell\left(z_i; \theta_{-j}^{[t]}\right), & t \neq t_{z_j}. \end{cases} \quad (5)$$

The parameter deviation $\mu^{[t]} := \theta^{[t]} - \theta_{-j}^{[t]}$ measures the cumulative influence of $z_j$ on model parameters. Before $z_j$

---

[1]At $t = t_{z_j}$, we replace $z_j$ with a zero-gradient placeholder and keep the same batch size and averaging denominator.

is encountered, both trajectories are identical, so $\mu^{[t]} = 0$ for all $t \leq t_{z_j}$. At step $t_{z_j}$, the initial deviation caused by excluding $z_j$ is:

$$\mu^{[t_{z_j}+1]} = \mu^{[t_{z_j}]} - \eta_{t_{z_j}} (\bar{g}^{[t_{z_j}]} - \bar{g}_{-j}^{[t_{z_j}]}) = -\frac{\eta_{t_{z_j}}}{|S_{t_{z_j}}|} \nabla_\theta \ell(z_j; \theta^{[t_{z_j}]}). \quad (6)$$

This initial deviation propagates through subsequent optimization steps. Following the influence function framework (Hara et al., 2019), the parameter gap evolves as:

$$\mu^{[t+1]} \approx \left(I - \eta_t H^{[t]}\right) \mu^{[t]}, H^{[t]} := \frac{1}{|S_t|} \sum_{i \in S_t} \nabla_\theta^2 \ell(z_i; \theta^{[t]}). \quad (7)$$

Define the influence propagator matrix at step $s$ as $P^{[s]} := I - \eta_s H^{[s]}$, and the cumulative propagator from step $a$ to step $b-1$ as: $P^{[a,b]} := \prod_{s=a}^{b-1} P^{[s]} = P^{[b-1]} P^{[b-2]} \cdots P^{[a]}$, where the product is ordered with later steps on the left. Unrolling from the initial deviation to the current step $\tau$, we obtain:

$$\mu^{[\tau]} = P^{[t_{z_j}+1,\tau]} \mu^{[t_{z_j}+1]} \approx -\frac{\eta_{t_{z_j}}}{|S_{t_{z_j}}|} P^{[t_{z_j}+1,\tau]} \nabla_\theta \ell(z_j; \theta^{[t_{z_j}]}). \quad (8)$$

Therefore, the model after removing the influence of $z_j$ is:

$$\begin{aligned} \hat{\theta}_{-j}^{[\tau]} &= \theta^{[\tau]} - \mu^{[\tau]} \\ &\approx \theta^{[\tau]} + \frac{\eta_{t_{z_j}}}{|S_{t_{z_j}}|} P^{[t_{z_j}+1,\tau]} \nabla_\theta \ell(z_j; \theta^{[t_{z_j}]}). \end{aligned} \quad (9)$$

This formulation enables *online* unlearning, as it requires only mini-batch gradients and local curvature information, without waiting for model convergence.

### 4.2. Mini-Batch Unlearning Strategy

The above sample-level influence tracking requires storing per-sample gradients and computing individual influence propagation, resulting in $O(Np)$ memory and $O(N)$ propagation operations, which are prohibitive for large models.

**Key Insight.** All data within an SGD mini-batch jointly induce a single parameter update, making their influences inherently entangled through the shared optimization step. This observation motivates treating mini-batches as atomic units for influence tracking and removal, which we term *mini-batch unlearning strategy*.

**Mini-Batch Unlearning Strategy.** To unlearn the target sample $z_j$, we remove the influence of the *SGD mini-batch* $S_{t_{z_j}}$ that contains $z_j$ and is used at training step $t_{z_j}$. We consider the counterfactual trajectory in which the parameter update induced by $S_{t_{z_j}}$ is *skipped*, i.e., $\theta_{-S_{t_{z_j}}}^{[t_{z_j}+1]} = \theta^{[t_{z_j}]}$. The initial deviation becomes:

$$\mu^{[t_{z_j}+1]} = \theta^{[t_{z_j}+1]} - \theta_{-S_{t_{z_j}}}^{[t_{z_j}+1]} = \theta^{[t_{z_j}+1]} - \theta^{[t_{z_j}]} = v^{[t_{z_j}]}, \quad (10)$$

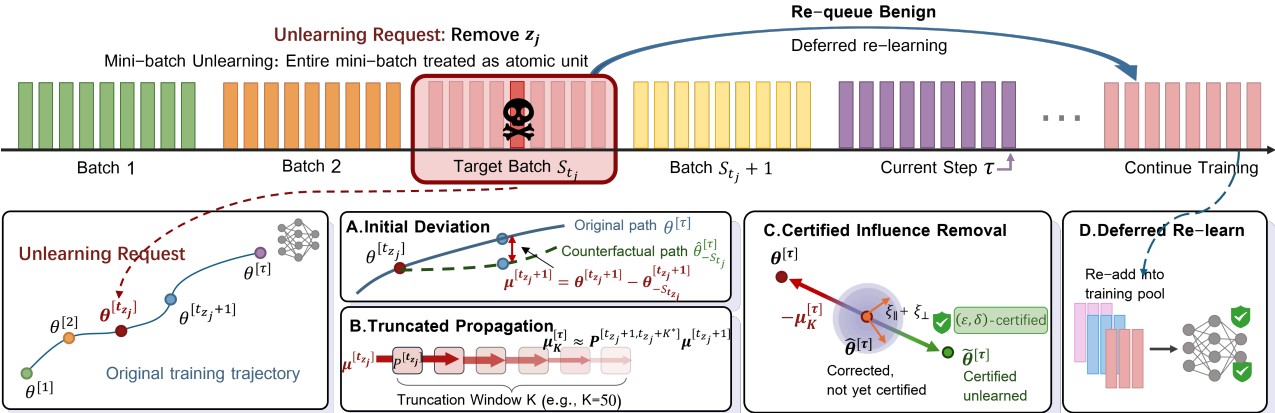

*Figure 1.* **Overview of LMCLEANER.** Upon an unlearning request for $z_j$, we: 1) treat target batch $S_{t_{z_j}}$ as an atomic unit, 2) compute influence propagation within a $K$-step window, and 3) inject noise for certified removal; 4) re-learn benign data in the target batch later.

where we define the SGD update vector at step $t$ as $v^{[t]} := \theta^{[t+1]} - \theta^{[t]} = -\eta_t \bar{g}^{[t]}$.

Propagating forward via Eq. (7), the cumulative deviation at step $\tau$ is:

$$\mu^{[\tau]} = P^{[t_{z_j}+1, \tau]} \cdot v^{[t_{z_j}]}. \quad (11)$$

Thus, the unlearned parameters are $\widehat{\theta}^{[\tau]}_{-S_{t_{z_j}}} = \theta^{[\tau]} - \mu^{[\tau]}$.

This mini-batch unlearning strategy reduces storage from $O(Np)$ (per-sample gradients) to $O(\frac{N}{B}p)$. We store parameter snapshots, and Hessian-vector products are computed on-demand via automatic differentiation, avoiding $O(p^2)$ Hessian storage. This strategy achieves a $512\times$ reduction from per-sample storage. For resource-constrained settings, sparse checkpointing every $C$ steps further reduces storage to $O(Tp/C)$ by reconstructing intermediate parameters on demand.

Since mini-batch unlearning removes the influence of the entire batch $S_{t_{z_j}}$, we re-add the contribution of benign data through *deferred re-learning*. Specifically, we restore the contribution of affected benign data to subsequent training, allowing them to contribute to future training steps. Our empirical analysis in Section 6.3 demonstrates that this compensation effectively preserves model utility, outperforming baselines.

### 4.3. Influence Propagation Truncation Mechanism

While mini-batch unlearning reduces storage costs, computing the full influence propagation allows for the most accurate removal but is computationally infeasible. The exact calculation requires $O(\tau - t_{z_j})$ Hessian-vector products, which is prohibitive when LLM training spans millions of steps. Furthermore, the reliance on first-order Taylor expansion implies that the influence estimate is mathematically only within a local temporal neighborhood, beyond which

linearization errors accumulate.

**Key Insight.** We observe that the influence of a data batch decays and diffuses over time, allowing the trajectory to be decomposed into a reliable *trust region* of $K$ steps and a *long-tail residual*. We therefore remove the deterministic component within the trust region (where local linearization is most accurate), and treat the remaining tail as a bounded residual that can be masked by the certification noise. Concretely, we propagate the deviation to $t_{\text{end}} = \min\{\tau, t_{z_j} + K + 1\}$ and then hold it constant; the resulting tail error is bounded in Appendix C.3.

**Truncated Propagation.** Let $K^\star := \min\{K, \tau - t_{z_j} - 1\}$ and $t_{\text{end}} := t_{z_j} + K^\star + 1$. Define the (regularized) local propagator $\tilde{P}^{[s]} := I - \eta_s \tilde{H}^{[s]}$ with $\tilde{H}^{[s]} := H^{[s]} + \lambda_{\text{reg}} I$, and the ordered product $\tilde{P}^{[a,b]} := \tilde{P}^{[b-1]} \ldots \tilde{P}^{[a]}$ (later steps on the left). We compute the held deviation as

$$\mu_K^{[\tau]} := \tilde{P}^{[t_{z_j}+1, t_{\text{end}}]} v^{[t_{z_j}]}, \quad (12)$$

and update the unlearned parameters by $\widehat{\theta}^{[\tau]} := \theta^{[\tau]} - \mu_K^{[\tau]}$. We do not explicitly compute the tail. Instead, its worst-case magnitude is absorbed into the public sensitivity bound used for noise calibration.

**Subspace-Aware Noise Injection.** To certify unlearning, we perturb the released parameters by Gaussian noise. Let $\Pi_k$ be a fixed, public projector onto a $k$-dimensional subspace (e.g., from a public pilot estimate or public randomness; Appendix C.5). We sample $\xi = \xi_\parallel + \xi_\perp$ with independent components

$$\xi_\parallel \sim \mathcal{N}(0, \sigma_\parallel^2 \Pi_k), \xi_\perp \sim \mathcal{N}(0, \sigma_\perp^2(I - \Pi_k)), \quad (13)$$

so that $\xi \sim \mathcal{N}(0, \sigma_\parallel^2 \Pi_k + \sigma_\perp^2(I - \Pi_k))$.

**Algorithm 1** LMCLEANER: Certified Online Unlearning
***
**Require:** Target batch step $t_{z_j}$; Current parameters $\theta^{[\tau]}$; Truncation window $K$; Privacy parameters $(\varepsilon, \delta)$; Training log $\mathcal{L} = \{\theta^{[t]}, S_t, \eta_t\}_{t=0}^{\tau}$; Public bound $\bar{\Delta}_{\text{cert}}(K)$; Concentration factor $\beta$; Regularization $\lambda_{\text{reg}} \geq 0$.

**Ensure:** Unlearned parameters $\widetilde{\theta}^{[\tau]}$
 1: **Phase 1: Compute Initial Deviation**
 2: $v \leftarrow \theta^{[t_{z_j}+1]} - \theta^{[t_{z_j}]}$
 3: **Phase 2: Influence Propagation Truncation**
 4: $K^{\star} \leftarrow \min\{K, \tau - t_{z_j} - 1\}$
 5: **for** $s = t_{z_j} + 1$ **to** $t_{z_j} + K^{\star}$ **do**
 6: $\quad Hv \leftarrow \text{COMPUTEHVP}(\theta^{[s]}, v)$
 7: $\quad Hv \leftarrow Hv + \lambda_{\text{reg}} \cdot v$ {Regularization for non-convexity}
 8: $\quad v \leftarrow v - \eta_s \cdot Hv$
 9: **end for**
10: **Phase 3: Certified Influence Removal**
11: $\mu_K^{[\tau]} \leftarrow v$
12: $\widehat{\theta}^{[\tau]} \leftarrow \theta^{[\tau]} - \mu_K^{[\tau]}$
13: $\sigma_{\parallel}, \sigma_{\perp} \leftarrow \text{CALIBRATENOISE}(\varepsilon, \delta, \bar{\Delta}_{\text{cert}}(K), \beta)$
14: $\xi \leftarrow \text{SAMPLENOISE}(\sigma_{\parallel}, \sigma_{\perp})$
15: $\widetilde{\theta}^{[\tau]} \leftarrow \widehat{\theta}^{[\tau]} + \xi$
16: **return** $\widetilde{\theta}^{[\tau]}$
***

The noise scales are calibrated from a *public* sensitivity upper bound $\bar{\Delta}_{\text{cert}}(K)$ (Section C.5) by applying the Gaussian mechanism to the two orthogonal blocks with budgets $(\varepsilon/2, \delta/2)$ each:

$$\sigma_{\parallel} = \frac{2\bar{\Delta}_{\text{cert}}(K)}{\varepsilon}\sqrt{2\log\left(\frac{2.5}{\delta}\right)}, \qquad (14)$$

$$\sigma_{\perp} = \frac{2\beta\bar{\Delta}_{\text{cert}}(K)}{\varepsilon}\sqrt{2\log\left(\frac{2.5}{\delta}\right)}. \qquad (15)$$

Here $\beta \in (0, 1]$ is the concentration factor. Algorithm 1 details this procedure.

**Selection of $K$.** The truncation window $K$ trades computation for utility. Each additional step in the window costs one Hessian–vector product, so the unlearning overhead scales as $O(K)$. Meanwhile, the tail residual decays exponentially with $K$ (Appendix C.3), reducing the total sensitivity and thus the certification noise. Since the Gaussian mechanism requires $\sigma_{\parallel}, \sigma_{\perp} = \Theta(\bar{\Delta}_{\text{cert}}(K))$, the utility loss from perturbation scales with $\bar{\Delta}_{\text{cert}}(K)^2$. In practice, we fix $K$ as a public hyperparameter and choose the smallest $K$ such that the tail term becomes negligible compared to the intrinsic linearization term (i.e., diminishing returns). Appendix C.6.1 provides an explicit approximation for $K$ by minimizing a public upper bound of $\bar{\Delta}_{\text{cert}}(K)$.

## 5. Theoretical Analysis

This section establishes guarantees for LMCLEANER. All assumptions and proofs are deferred to Appendix C.

### 5.1. Approximation Error Bound.

Our goal is to show that the parameters $\widehat{\theta}^{[\tau]}$ produced by LMCLEANER are close to the *counterfactual model*, obtained by skipping the update at the target step and continuing training identically thereafter.

Let $t_0$ denote the training step where the target batch was used, $K^{\star} = \min\{K, \tau - t_0 - 1\}$, and $t_{\text{end}} := \min\{\tau, t_0 + K + 1\}$. Skipping the step-$t_0$ update induces an initial deviation $\mu_0 := \theta^{[t_0+1]} - \theta^{[t_0]} = v^{[t_0]}$. LMCLEANER propagates this deviation using the local linearization $\mu_{\text{lin}}^{[t+1]} = (I - \eta_t \tilde{H}^{[t]})\mu_{\text{lin}}^{[t]}$ for $K^{\star}$ steps, and then holds it constant, i.e., $\mu_K^{[\tau]} := \mu_{\text{lin}}^{[t_{\text{end}}]}$ and $\widehat{\theta}^{[\tau]} := \theta^{[\tau]} - \mu_K^{[\tau]}$.

The approximation error has two components. Within the truncation window, first-order Taylor expansion incurs an *in-window linearization error* $\Delta_{\text{lin}}(K^{\star})$. Beyond the window, holding the deviation introduces a *tail residual* $R(K)$. Under contractive SGD dynamics (Appendix C.2), the tail residual decays exponentially with $K$, making truncation computationally efficient and theoretically sound.

**Theorem 5.1** (Deterministic approximation error). *Under Assumptions C.2–C.3 (Appendix C), for any $\tau \geq t_0 + 1$,*

$$\|\widehat{\theta}^{[\tau]} - \theta_{-S_{t_0}}^{[\tau]}\|_2 \leq \Delta_{\text{cert}}(K) := \Delta_{\text{lin}}(K^{\star}) + R(K), \quad (16)$$

*where $\Delta_{\text{lin}}(K^{\star})$ is the in-window linearization error and $R(K)$ is the tail residual decaying exponentially in $K$.*

The bound reveals a trade-off: increasing $K$ reduces the tail residual exponentially, while the linearization error grows at most linearly and is bounded under contraction. In practice, a moderate window ($K \approx 50$) suffices to make $R(K)$ negligible, as validated in Section 6. See Appendix C.3 for explicit formulas.

### 5.2. Certified Unlearning

To achieve certified indistinguishability, we inject calibrated Gaussian noise to mask the residual gap. We observe that the approximation error concentrates in a low-dimensional subspace spanned by recent gradient directions. By injecting stronger noise only in this subspace and weaker noise elsewhere, we achieve the same privacy guarantee with less overall perturbation.

Let $\mathcal{M}(\theta) := \theta + \xi$ where $\xi$ is Gaussian with covariance $\sigma_{\parallel}^2 \Pi_k + \sigma_{\perp}^2 (I - \Pi_k)$, and the reference retraining output is $\mathcal{M}(\theta_{-S_{t_0}}^{[\tau]})$. The noise scales must be calibrated from the error bound $\Delta_{\text{cert}}(K)$ in Theorem 5.1. However, $\Delta_{\text{cert}}(K)$

*Table 1.* Comparison of Unlearning Methods (Capability, Total Overhead, and FLOPs Magnitudes per deletion request).

| Method | Capability | | | Total Comput. | Storage | Est. | Data Need for Unlearning | |
|---|---|---|---|---|---|---|---|---|
| | Online | Sample-targeted | Certified | Complexity | Overhead | FLOPs | Forget | Retain |
| Retrain | × | ✓ | Exact | $O(E_{\text{train}}|D \setminus U|Lp)$ | $O(|D|d)$ | $\sim 10^{21}$ | – | Required |
| GradDiff (Maini et al., 2024) | × | × | × | $O(E_u|D|Lp)$ | $O(|D|d)$ | $\sim 10^{20}$ | Required | Required |
| NPO (Zhang et al., 2024) | × | × | × | $O(E_u|D_f|Lp)$ | $O(|D_f|d+p)$ | $\sim 10^{19}$ | Required | – |
| UNDIAL (Dong et al., 2025) | × | × | × | $O(E_u|D_f|Lp)$ | $O(|D|d+p)$ | $\sim 10^{19}$ | Required | – |
| PDU (Entesari et al., 2025) | × | × | × | $O(E_u|D|Lp)$ | $O(|D|d)$ | $\sim 10^{20}$ | Required | Required |
| HFR (Qiao et al., 2025) | ✓ | ✓ | ✓ | $O(E_{\text{train}}|D|^2 Lp/B + |U|p)$ | $O(|D|p)$ | $\sim 10^{23}$ | – | – |
| **LMCLEANER** | ✓ | ✓ | ✓ | **$O(KBLp)$** | **$O(Tp)$** | $\sim \mathbf{10^{17}}$ | – | – |

$p = 1.2 \times 10^9$ params, $d$ (sample size) $\approx L$ (seq. length)$= 1024$, $|D|$ (training set size)$= 10^6$, with $|D_f| = 0.1|D|$ and $|D_r| = 0.9|D|$; $E_{\text{train}}$ (training epochs)$= 100$, $E_u$ (unlearning epochs)$= 10$, $B = 512$, $T \approx 1.95 \times 10^5$, $K = 50$, and $|U|$ is the deletion request size.

depends on $\|\mu_0\|_2$, which varies with the specific batch being unlearned. To ensure data-independent calibration, we use gradient clipping with threshold $C$ to obtain a public upper bound $\|\mu_0\|_2 \le \eta_{\max}C =: G_{\max}$, and define $\bar{\Delta}_{\text{cert}}(K)$ by substituting $G_{\max}$ for $\|\mu_0\|_2$ in $\Delta_{\text{cert}}(K)$.

**Theorem 5.2** (Certified unlearning via subspace-aware Gaussian noise). *Assume (i) $\Pi_k$ is public/data-independent and (ii) the deterministic mean shift $e := \widehat{\theta}^{[\tau]} - \theta^{[\tau]}_{-S_{t_0}}$ satisfies $\|(I - \Pi_k)e\|_2 \le \beta\|e\|_2$ for a public $\beta \in (0,1]$. Release*

$$\widetilde{\theta}^{[\tau]} := \mathcal{M}(\widehat{\theta}^{[\tau]}) = \widehat{\theta}^{[\tau]} + \xi, \xi \sim \mathcal{N}\big(0, \sigma_\|^2\Pi_k + \sigma_\perp^2(I - \Pi_k)\big),$$

*where $(\sigma_\|, \sigma_\perp)$ are set as in Eq. (14). Then $\widetilde{\theta}^{[\tau]}$ is $(\varepsilon, \delta)$-indistinguishable from the noisy batch-skip reference $\mathcal{M}(\theta^{[\tau]}_{-S_{t_0}})$. This establishes certified indistinguishability for the atomic mini-batch removal step.*

See proof in Appendix C.6.

## 6. Experiments

We evaluate LMCLEANER on the TOFU benchmark (Maini et al., 2024) using OpenUnlearning (Dorna et al., 2025).

### 6.1. Experimental Setup

**Dataset and Models.** TOFU contains 200 fictitious authors with 20 QA pairs each; we use the standard 10% forget split ($|D_f|=400$). We fine-tune LLaMA-3.2-3B, LLaMA-3.2-1B, and GPT-2 for five epochs, saving checkpoints at each epoch to simulate online unlearning requests at different training stages. All methods start from identical checkpoints for fair comparison. By default, truncation window $K=50$ (ablation in Section 6.5).

**Baselines.** We compare against six methods: GradDiff (Maini et al., 2024), NPO (Zhang et al., 2024), UNDIAL (Dong et al., 2025), PDU (Entesari et al., 2025), HFR (Qiao et al., 2025), and Retrain. We do not run HFR in the LLaMA-3.2-1B experiments and only include it in the capability/overhead analysis, due to its prohibitive training-time precomputation at LLM scale (see Table 1).

**Evaluation protocol.** To evaluate online unlearning across different training stages, we construct *independent* unlearning scenarios at each epoch boundary. For each $e \in \{1, 2, 3, 4, 5\}$, we first fine-tune normally for $e$ epochs to obtain checkpoint $\theta^{[e]}$, then issue an unlearning request for the forget set $D_f$, apply each method, and resume training under the same remaining schedule until the final epoch. The final model is evaluated on forget quality, utility, and privacy metrics. Scenarios at different epochs are independent and not cumulative, enabling controlled comparisons across progressively more challenging online-unlearning settings.

This protocol naturally yields two complementary evaluation regimes. *Epoch 1* matches the single-pass ordering assumed in Section 3, since each training sample has appeared at most once before the request. Theorems 5.1–5.2 apply directly to each affected sample occurrence or containing mini-batch. *Epochs 2–5* further evaluate the multi-occurrence setting where each target sample may have contributed across multiple epochs before unlearning.

### 6.2. Capability and Efficiency Comparison

Table 1 summarizes the capability and overhead of representative unlearning methods.

Fine-tuning-based approaches (GradDiff, NPO) incur substantial compute overhead ($\sim 10^{19}$ FLOPs) due to post-training optimization over the forget set, and require access to raw forget data at deletion time. HFR provides certified online deletion by shifting the cost to training time, but requires intractable pre-computation complexity $O(E_{\text{train}}|D|^2 Lp/B)$ and prohibitive storage $O(|D|p)$, which do not scale to LLM settings. In contrast, LMCLEANER performs unlearning using only a short window, yielding orders-of-magnitude lower per-request overhead.

We highlight three practical advantages of LMCLEANER. 1) Sample-targeted Unlearning. Real-world unlearning requests are targeted at specific training data and arrive one by one. Fine-tuning methods struggle with this granularity, as optimizing on a single sample may produce insufficient

*Table 2.* **Comprehensive evaluation on TOFU benchmark at Epoch 5 (LLaMA-3.2-1B).** ↑: higher is better; ↓: lower is better; `forget_truth` closer to 0.5 is better. Δ**Retrain**: absolute aggregate-score gap to the checkpoint-conditioned Retrain reference; lower is better. The  Retrain  row serves as the gold standard.

| Method | Model Utility | | | | | | Forget Quality | | | | Distrib. |
|---|---|---|---|---|---|---|---|---|---|---|---|
| | utility↑ | ret_RG↑ | ret_Prob↑ | ret_Truth↑ | ra_RG↑ | wf_RG↑ | fg_truth→0.5 | fg_RG↓ | fg_Prob↓ | extract↓ | ΔRetrain↓ |
| Retrain | 0.430 | 0.392 | 0.264 | 0.274 | 0.888 | 0.873 | 0.721 | 0.392 | 0.211 | 0.064 | — |
| GradDiff | 0.270 | 0.180 | 0.160 | 0.305 | 0.163 | 0.446 | 0.003 | **0.011** | **0.000** | **0.033** | 2.938 |
| NPO | 0.416 | 0.339 | 0.234 | 0.296 | 0.725 | 0.757 | 0.688 | 0.195 | 0.025 | 0.049 | 0.785 |
| PDU | 0.415 | 0.378 | 0.228 | 0.266 | **0.872** | 0.843 | 0.730 | 0.402 | 0.222 | 0.078 | 0.075 |
| UNDIAL | 0.395 | 0.380 | 0.177 | 0.278 | 0.869 | 0.841 | 0.726 | 0.359 | 0.161 | 0.057 | 0.266 |
| **LMCLEANER** | **0.429** | **0.387** | **0.244** | **0.312** | 0.857 | **0.854** | **0.645** | 0.412 | 0.282 | 0.076 | **0.011** |

parameter shift or unstable updates. LMCLEANER supports sample-level unlearning by identifying its containing mini-batch update as the atomic unit, then compensating benign samples via deferred re-learning, yielding an effective sample-targeted behavior. 2) Training-Data-Free. Fine-tuning methods require access to raw data during deletion, creating storage burdens $O(|D||d|)$. Moreover, this conflicts with data protection regulations that mandate prompt data deletion (Voigt & Von dem Bussche, 2024). HFR requires storing per-sample influence vectors during training, incurring $O(|D|p)$ storage. For LLaMA-3.2-1B with $|D| = 10^6$ samples, these amounts exceed practical limits. LMCLEANER does not require raw forget samples at deletion time: the initial deletion vector $v^{[t_0]}$ is obtained from the stored trajectory log. 3) Certified Guarantee. Heuristic methods such as GradDiff and NPO do not provide formal guarantees that forgotten data cannot be recovered. In contrast, LMCLEANER provides a formal $(\varepsilon, \delta)$-certified unlearning guarantee, and HFR does as well, at substantially higher training-time cost.

### 6.3. Forget Quality and Model Utility

An effective unlearning method should remove the influence of target data while preserving performance on retained knowledge. We evaluate unlearning effectiveness from two aspects: *forget quality*, measuring how completely the model loses knowledge of target data; and *model utility*, measuring performance preservation on retained knowledge.

Figure 2 presents performance across five training epochs, and Table 2 provides comprehensive metrics at Epoch 5.

**Results Analysis.** Figure 2 demonstrates that LMCLEANER achieves the near-optimal trade-off between forgetting effectiveness and utility preservation across both models and all training epochs. On LLaMA-3.2-1B, LMCLEANER achieves the best forget truth among non-collapsing baselines; on GPT-2, it remains comparable to Retrain while maintaining the strongest utility. While GradDiff exhibits lower forget truth ratios, this comes at the

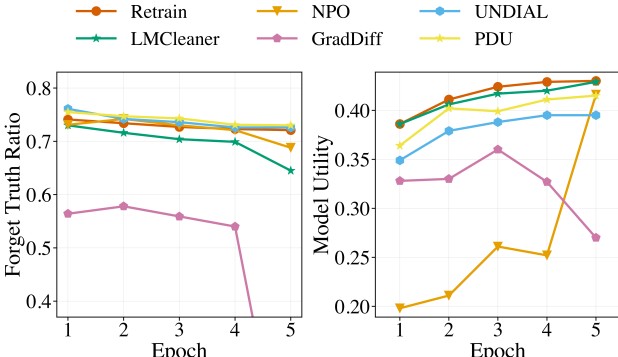

*(a)* Forget truth ratio and model utility of LLaMA-3.2-1B

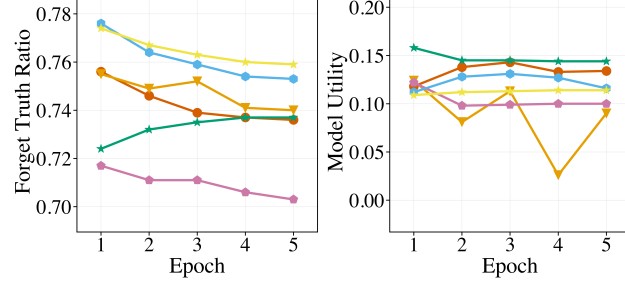

*(b)* Forget truth ratio and model utility of GPT-2

*Figure 2.* **Performance comparison across training epochs on TOFU benchmark.** LMCLEANER consistently achieves superior or comparable forget quality while maintaining model utility across both model scales.

cost of catastrophic utility degradation (dropping to 0.27 on LLaMA-3.2-1B by Epoch 5), rendering the model unusable. In contrast, LMCLEANER achieves forgetting while preserving full model capability. Second, LMCLEANER's utility curves closely track the Retrain baseline throughout training on both models. It matches Retrain's performance on LLaMA-3.2-1B and even slightly exceeds it on GPT-2, confirming that removing influence does not disrupt the learned capability. Third, LMCLEANER demonstrates superior stability across varying scales and stages. Baseline methods exhibit severe volatility: NPO suffers a catastrophic col-

*Table 3.* **LLaMA-3.2-3B results (TOFU, Epoch 1).**

| Method | Utility↑ | fg_truth→0.5 | MIA(min_k)→0.5 | MIA(loss)→0.5 |
|---|---|---|---|---|
| Retrain | 0.320 | 0.746 | 0.320 | 0.319 |
| GradDiff | 0.276 | **0.523** | 0.046 | 0.052 |
| NPO | 0.250 | 0.703 | 0.012 | 0.016 |
| PDU | **0.439** | 0.750 | 0.287 | 0.252 |
| UNDIAL | 0.311 | 0.752 | 0.266 | 0.203 |
| **LMCLEANER** | 0.431 | 0.739 | **0.332** | **0.340** |

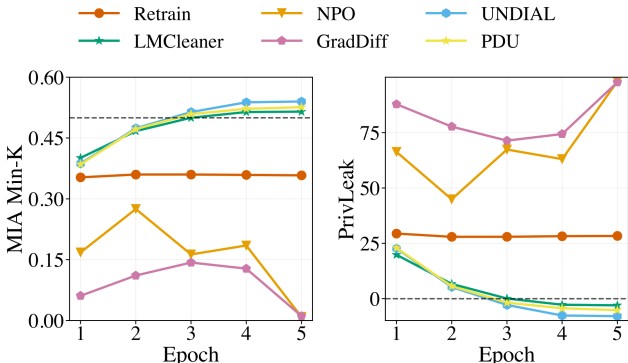

*Figure 3.* **Privacy evaluation across training epochs.** Left: MIA accuracy; LMCLEANER converges to the random-guess ideal (0.5). Right: PrivLeak; LMCLEANER approaches 0, indicating minimal privacy leakage compared to baselines.

lapse at Epoch 4 on GPT-2, GradDiff shows progressive decay, and PDU/UNDIAL show limited improvement in terms of fg_truth beyond Retrain. These results validate the fundamental advantage of online unlearning: removing data influence early enables cleaner forgetting without the instabilities inherent to post-training approaches.

Table 2 further quantifies these findings through the $\Delta$ Retrain metric, $\Delta$Retrain $= |\sum_i m_i^{\text{method}} - \sum_i m_i^{\text{Retrain}}|$, where $m_i$ denotes the $i$-th metric. LMCLEANER achieves the smallest gap, significantly smaller than all baselines, confirming it yields the closest distributional approximation to the retrained gold standard.

We provide additional visualizations in Appendix E, including utility-forget trade-off trajectories (Figure 4) that demonstrate LMCLEANER's stability across epochs, and radar charts (Figure 5) showing comprehensive multi-metric comparisons.

To validate scalability, we additionally evaluate on LLaMA-3.2-3B at Epoch 1 (Table 3). LMCLEANER attains utility competitive with the best-performing baseline, while uniquely achieving MIA scores closest to the ideal 0.5 baseline under both `min_k` and `loss` criteria. Although GradDiff's fg_truth (0.523) appears nominally closest to 0.5, its near-zero MIA scores (0.046/0.052) indicate that this is the byproduct of privacy collapse rather than principled forgetting. Among methods that simultaneously preserve utility and privacy, LMCLEANER delivers the most balanced trade-off, confirming that it scales effectively to larger models.

### 6.4. Privacy Evaluation

Beyond forget quality and utility, a key requirement for unlearning is *privacy*: an adversary should not be able to distinguish whether a sample was in the original training set by querying the unlearned model. This is particularly important for compliance with data protection regulations such as GDPR's Right to be Forgotten, where the goal is not merely to degrade model performance on forgotten data, but to ensure that no detectable trace of that data remains.

Following OpenUnlearning, we evaluate privacy using two metrics: 1) *Membership Inference Attacks (MIA)*, which measures attack success rate. 0.5 indicates ideal privacy,

which shows random guessing. 2) *PrivLeak*, which quantifies the privacy gap from the theoretical ideal. A PrivLeak of 0 indicates perfect indistinguishability.

Figure 3 presents the privacy metrics across all training epochs. LMCLEANER consistently achieves the best privacy protection. At Epoch 3, LMCLEANER attains an MIA accuracy of 0.5 and a PrivLeak of 0.05, effectively rendering forgotten samples indistinguishable from never-seen data.

**Results Analysis.** The baseline methods exhibit two distinct failure modes. GradDiff and NPO show severe *over-unlearning*: their MIA scores approach 0 rather than 0.5, and PrivLeak spikes significantly. This result arises because aggressive gradient modifications destroy model coherence, making the forget set trivially distinguishable via reverse leakage, not due to privacy protection, but due to model collapse. Conversely, PDU and UNDIAL show mild *under-unlearning*: MIA scores hover above 0.5, indicating that residual information about the forget set remains detectable. Notably, LMCLEANER is closer to the random-guessing MIA value of 0.5 than the Retrain baseline in this empirical attack evaluation.

### 6.5. Ablation Study

#### 6.5.1. EFFECT OF TRUNCATION WINDOW $K$

The truncation window $K$ controls the trade-off between computational cost and influence estimation accuracy. Larger $K$ captures more of the influence propagation chain but requires proportionally more Hessian-vector products. We evaluate $K \in \{50, 100, 500, 1000, 1250\}$ across training epochs. Table 4 reports results at Epoch 5.

**Results Analysis.** Table 4 reveals two important insights. First, LMCLEANER remains remarkably stable from $K = 50$ to $K = 1000$. Utility, forget quality, and MIA vary

*Table 4.* **Ablation on truncation window** $K$ (Epoch 5, LLaMA-3.2-1B).

| K | Utility↑ | Forget Truth→ 0.5 | MIA → 0.5 | Relative Cost |
|---|---|---|---|---|
| **50** | 0.429 | **0.645** | **0.515** | 1× |
| 100 | **0.430** | **0.645** | 0.516 | 2× |
| 500 | 0.428 | 0.650 | 0.516 | 10× |
| 1000 | 0.428 | 0.650 | 0.516 | 20× |
| 1250 | 0.429 | 0.650 | 0.599 | 25× |

only marginally despite a $20\times$ increase in computational cost. This empirically validates the analysis in Section 4.3: the influence tail decays rapidly, and most meaningful influence is concentrated within a short local propagation window. Second, excessively large windows can harm approximation quality. At $K = 1250$, MIA degrades to 0.599, indicating that long-horizon linearization errors accumulate faster than the residual influence decreases. This directly matches the trade-off predicted by Theorem 5.1: increasing $K$ reduces the tail residual but also increases sensitivity to approximation error. In practice, $K = 50$ already achieves near-optimal performance with minimal computational overhead.

### 6.5.2. EFFECT OF BATCH GRANULARITY $B$

A central design choice in LMCLEANER is treating mini-batch updates as atomic units for online unlearning. Larger batch sizes reduce storage and propagation overhead, but may introduce more collateral forgetting because more benign samples are removed together with the target sample. To quantify this trade-off, we vary the mini-batch size $B \in \{8, 16, 32, 64, 128, 256\}$.

*Table 5.* **Effect of mini-batch size** $B$ **on unlearning performance.** As $B$ increases, forgetting improves gradually while utility degrades smoothly rather than catastrophically.

| B | Utility↑ | FG Prob↓ | MIA→ 0.5 |
|---|---|---|---|
| 8 | 0.388 | 0.282 | 0.575 |
| 16 | 0.386 | 0.275 | 0.556 |
| 32 | 0.383 | 0.268 | 0.536 |
| 64 | 0.379 | 0.254 | 0.509 |
| 128 | 0.365 | 0.226 | 0.461 |
| 256 | 0.349 | 0.201 | 0.452 |

**Results Analysis.** Table 5 demonstrates that the degradation induced by batch-level unlearning is gradual rather than catastrophic. As $B$ increases, forgetting becomes stronger because each affected batch contains more target-associated updates, while utility decreases smoothly due to increased benign collateral removal. Importantly, the model remains stable even under aggressive settings. At $B = 64$, MIA reaches 0.509, nearly matching the ideal random-guessing baseline, while utility remains stable. Even at $B = 256$,

utility decreases by only about 10% without collapse.

These findings provide empirical support for the deferred re-learning mechanism: although removing larger batches introduces collateral damage, the benign samples reintroduced during subsequent training effectively absorb most of the trajectory drift. The default setting $B = 64$ achieves the best overall balance between utility, forget quality, and privacy.

## 7. Conclusion

We presented LMCLEANER, a novel online unlearning framework for large language models that removes data influence *during* training rather than after convergence. Our key insight is that influence propagation can be truncated within a bounded window and the residual masked by calibrated noise, enabling efficient removal with $(\varepsilon, \delta)$-certified guarantees. Experiments on TOFU demonstrate that LMCLEANER achieves a strong utility–forgetting trade-off, closely tracks the Retrain reference on model utility, and substantially reduces per-request computation compared with fine-tuning-based baselines.

## Acknowledgements

We thank the anonymous reviewers for their constructive feedback. This work was supported in part by Hong Kong Research Grants Council (RGC) under Grants RFS2425-1S01, RFS2122-1S04, R1012-21, C6015-23G, and CRS_HKUST601/24.

## Impact Statement

LMCLEANER enables practical compliance with data protection regulations such as GDPR's "right to be forgotten" for large language models. By providing certified unlearning guarantees with tractable computational costs, our method can help responsible AI deployment across organizations with limited computational resources.

However, the ability to efficiently remove specific training data could potentially be misused by malicious adversaries to selectively eliminate safety-relevant training data, degrading model alignment. We recommend that unlearning operations be logged and audited, and that critical safety data be protected from removal requests.

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

# A. Extended Related Work

Machine unlearning methods can be broadly classified into exact unlearning and approximate unlearning (Xu et al., 2024).

**Exact Unlearning**   Exact unlearning aims to achieve a model state identical to one retrained from scratch on the remaining dataset. The foundational approach is SISA (Bourtoule et al., 2021), which divides training data into disjoint shards, trains independent sub-models on each, and then aggregates predictions. When data needs to be forgotten, only the sub-models containing that specific data require retraining, significantly reducing computational overhead compared to full retraining.

Building on SISA, various domain-specific adaptations have emerged. DC-k-means (Ginart et al., 2019) applies exact unlearning to k-means clustering by maintaining cluster statistics that can be efficiently updated upon data removal. For tree-based models, DaRE (Brophy & Lowd, 2021) and HedgeCut (Schelter et al., 2021) enable efficient removal from random forests, leveraging the inherent modularity of ensemble methods. Graph-based applications include GraphEraser (Chen et al., 2022b) for graph neural networks and RecEraser (Chen et al., 2022a) for recommendation systems, both exploiting structural properties to minimize recomputation. SISA-style partitioning reduces the effective training data for each sub-model and therefore degrades model quality. This makes it unsuitable for billion-parameter models that rely on large, unified training corpora. LMEraser (Xu et al., 2025b) employs a prompt tuning architecture that confines data partitioning to the prompt training stage, preserving the model backbone and enabling precise unlearning to scale to large language models.

Despite these advances, exact unlearning methods face fundamental scalability limitations. The computational cost grows prohibitively with model size and dataset complexity.

**Approximate Unlearning**   Approximate unlearning methods improve efficiency by reducing the influence of forgotten data, rather than exactly removing it.

Early approximate unlearning methods (Guo et al., 2020; Sekhari et al., 2021; Suriyakumar & Wilson, 2022) utilize influence functions to quantify how target training data affect model parameters. These methods estimate parameter changes that would result from removing specific data points, thus avoiding expensive retraining. Guo *et al.* (Guo et al., 2020) provided certified removal guarantees for convex models, while Sekhari *et al.* (Sekhari et al., 2021) extended the framework to more general settings. However, influence function-based approaches face significant challenges in deep learning contexts. Computing exact influence requires expensive Hessian inverses, and first-order approximations often become unreliable for highly non-convex objectives (Basu et al., 2021). These limitations have motivated more direct gradient-based interventions.

Gradient Ascent (GA) methods (Wu et al., 2020; Neel et al., 2021) perform gradient ascent on the loss function with respect to the forget set, effectively minimizing the model's ability to correctly predict on unwanted data. DeltaGrad (Wu et al., 2020) formalizes this approach, showing that gradient ascent can approximate the effect of data removal under certain conditions. While computationally efficient, these methods lack formal certified guarantees and can significantly degrade general model performance (Ilharco et al., 2023).

The scale of modern large language models poses unique challenges that traditional unlearning methods cannot address. This has motivated a surge of LLM-specific unlearning approaches, primarily based on fine-tuning, knowledge editing, or output manipulation. 1) Fine-tuning-based methods adapt the pre-trained model through additional optimization to suppress knowledge of forgotten data. Negative Preference Optimization (NPO) (Zhang et al., 2024) treats forgotten data as negative preferences within the DPO framework (Rafailov et al., 2023). SCRUB (Kurmanji et al., 2023) combines gradient ascent on forget data with knowledge distillation from a retain-set teacher. Fan et al. (2025) propose simplified fine-tuning objectives that achieve competitive results. While these methods are computationally tractable for LLMs, they operate heuristically without formal guarantees on the completeness of data influence removal. 2) Knowledge editing targets specific information encoded within model parameters. Task Vectors (Ilharco et al., 2023) subtract task-specific weight differences from model weights, effectively removing learned capabilities. Localization-based editing methods (Meng et al., 2022; Wu et al., 2023; Wei et al., 2024) identify and modify specific model components (layers, neurons, attention heads) responsible for encoding target knowledge. ROME (Meng et al., 2022) and MEMIT (Meng et al., 2023) demonstrate precise fact editing in language models by targeting specific layers and rank-one updates. However, these approaches require careful knowledge localization and may struggle with distributed representations. Recent work has expanded editing to broader knowledge categories. Knowledge Neurons (Jang et al., 2023) identifies and modifies neurons responsible for specific facts, while RWKU (Jin et al., 2024) provides unlearning for factual knowledge through targeted parameter updates. Sepahvand *et al.* (Sepahvand et al., 2025) develop selective editing techniques that preserve related knowledge while removing specific information. 3) Output filtering modifies model behavior at inference time without altering trained parameters. Corrupted prompts (Liu et al., 2024),

soft prompting (Bhaila et al., 2025), and in-context unlearning (Pawelczyk et al., 2024) guide outputs away from forgotten content. While highly efficient, these methods face a fundamental limitation: models retain internal representations of forgotten information, merely preventing explicit expression. This raises serious concerns about unlearning completeness and potential circumvention through extraction attacks.

The growth of LLM unlearning has motivated evaluation frameworks. TOFU (Maini et al., 2024) provides a benchmark with fictitious author profiles for controlled evaluation. MUSE (Shi et al., 2025) offers six-dimensional evaluation covering memorization, privacy, and utility. WMDP (Li et al., 2024) focuses on unlearning hazardous knowledge. These benchmarks reveal that existing methods often fail to completely remove target knowledge while preserving model utility. However, they generally overlook the computational and storage overhead of unlearning methods, leaving their practical scalability unassessed.

## B. Batch-Level Unlearning Derivation

This appendix provides the detailed derivation of the mini-batch unlearning formulation presented in Section 4.2. We show how removing an entire mini-batch reduces to tracking a single parameter deviation that propagates through subsequent optimization steps.

**Problem Setup**    Consider a mini-batch $S_{t_{z_j}}$ used for training at step $t_{z_j}$, which is now requested for unlearning at current step $\tau > t_{z_j}$. Our goal is to compute the parameter deviation $\mu^{[\tau]} := \theta^{[\tau]} - \theta^{[\tau]}_{-S_{t_{z_j}}}$, where $\theta^{[\tau]}$ denotes the actual parameters and that $\theta^{[\tau]}_{-S_{t_{z_j}}}$ denotes the hypothetical parameters that would have been obtained if batch $S_{t_{z_j}}$ had never been used.

We assume each training data appears at most once during training, which is standard in large-scale LLM pre-training where the dataset is traversed in a single epoch.

**Initial Deviation**    The actual trajectory with $S_{t_{z_j}}$ is:

$$\theta^{[t+1]} = \theta^{[t]} - \eta_t \bar{g}^{[t]}, \quad \bar{g}^{[t]} = \frac{1}{|S_t|} \sum_{i \in S_t} \nabla_\theta \ell(z_i; \theta^{[t]}), \tag{17}$$

where training proceeds normally through all steps including $t_{z_j}$.

The counterfactual trajectory without $S_{t_{z_j}}$ is:

$$\theta^{[t+1]}_{-S_{t_{z_j}}} = \begin{cases} \theta^{[t]}_{-S_{t_{z_j}}}, & t = t_{z_j} \text{ (step skipped)} \\ \theta^{[t]}_{-S_{t_{z_j}}} - \eta_t \bar{g}^{[t]}_{-S_{t_{z_j}}}, & t \neq t_{z_j} \end{cases} \tag{18}$$

where $\bar{g}^{[t]}_{-S_{t_{z_j}}} = \frac{1}{|S_t|} \sum_{i \in S_t} \nabla_\theta \ell(z_i; \theta^{[t]}_{-S_{t_{z_j}}})$ is the gradient at the counterfactual parameters.

Before step $t_{z_j}$, both trajectories are identical since $S_{t_{z_j}}$ has not yet been encountered. Thus, we have

$$\theta^{[t]} = \theta^{[t]}_{-S_{t_{z_j}}}, \quad \forall t \leq t_{z_j}. \tag{19}$$

Note that at step $t_{z_j}$, the trajectories diverge. In the actual trajectory, we apply the gradient update. In the counterfactual, we skip this step, which is:

$$\theta^{[t_{z_j}+1]} = \theta^{[t_{z_j}]} - \eta_{t_{z_j}} \bar{g}^{[t_{z_j}]}, \tag{20}$$

$$\theta^{[t_{z_j}+1]}_{-S_{t_{z_j}}} = \theta^{[t_{z_j}]}_{-S_{t_{z_j}}} = \theta^{[t_{z_j}]}. \tag{21}$$

The initial deviation is therefore:

$$\mu^{[t_{z_j}+1]} := \theta^{[t_{z_j}+1]} - \theta^{[t_{z_j}+1]}_{-S_{t_{z_j}}} = \theta^{[t_{z_j}+1]} - \theta^{[t_{z_j}]} = -\eta_{t_{z_j}} \bar{g}^{[t_{z_j}]} = v^{[t_{z_j}]}, \tag{22}$$

where $v^{[t_{z_j}]} := \theta^{[t_{z_j}+1]} - \theta^{[t_{z_j}]}$ is the parameter update at step $t_{z_j}$, consistent with the notation in Section 4.2.

**Deviation Propagation via Taylor Expansion**    For steps $t > t_{z_j}$, the deviation $\mu^{[t]} := \theta^{[t]} - \theta^{[t]}_{-S_{t_{z_j}}}$ evolves as:

$$\mu^{[t+1]} = \theta^{[t+1]} - \theta^{[t+1]}_{-S_{t_{z_j}}} = \left(\theta^{[t]} - \eta_t \bar{g}^{[t]}\right) - \left(\theta^{[t]}_{-S_{t_{z_j}}} - \eta_t \bar{g}^{[t]}_{-S_{t_{z_j}}}\right) = \mu^{[t]} - \eta_t \left(\bar{g}^{[t]} - \bar{g}^{[t]}_{-S_{t_{z_j}}}\right). \tag{23}$$

To relate $\bar{g}^{[t]}_{-S_{t_{z_j}}}$ to $\bar{g}^{[t]}$, we apply a first-order Taylor expansion around $\theta^{[t]}$:

$$\bar{g}^{[t]}_{-S_{t_{z_j}}} = \frac{1}{|S_t|} \sum_{i \in S_t} \nabla_\theta \ell(z_i; \theta^{[t]}_{-S_{t_{z_j}}}) \approx \frac{1}{|S_t|} \sum_{i \in S_t} \left[\nabla_\theta \ell(z_i; \theta^{[t]}) + \nabla^2_\theta \ell(z_i; \theta^{[t]}) \cdot (\theta^{[t]}_{-S_{t_{z_j}}} - \theta^{[t]})\right] = \bar{g}^{[t]} - H^{[t]} \mu^{[t]}, \tag{24}$$

where $H^{[t]} := \frac{1}{|S_t|} \sum_{i \in S_t} \nabla^2_\theta \ell(z_i; \theta^{[t]})$ is the mini-batch Hessian.

Substituting Eq. (24) into Eq. (23):

$$\mu^{[t+1]} = \mu^{[t]} - \eta_t\big(\bar{g}^{[t]} - (\bar{g}^{[t]} - H^{[t]}\mu^{[t]})\big) = \mu^{[t]} - \eta_t H^{[t]}\mu^{[t]} = (I - \eta_t H^{[t]})\mu^{[t]} = P^{[t]}\mu^{[t]}, \tag{25}$$

where $P^{[t]} := I - \eta_t H^{[t]}$ is the influence propagator defined in Section 4.1.

**Cumulative Deviation**   Unrolling the recursion in Eq. (25) from step $t_{z_j} + 1$ to $\tau$:

$$\mu^{[\tau]} = P^{[\tau-1]}\mu^{[\tau-1]} = P^{[\tau-1]}P^{[\tau-2]}\mu^{[\tau-2]} = P^{[\tau-1]}P^{[\tau-2]}\cdots P^{[t_{z_j}+1]}\mu^{[t_{z_j}+1]} \tag{26}$$

$$= P^{[t_{z_j}+1,\tau]} \cdot \mu^{[t_{z_j}+1]}, \tag{27}$$

where $P^{[a,b]} := \prod_{s=a}^{b-1} P^{[s]} = P^{[b-1]}P^{[b-2]}\cdots P^{[a]}$ is the cumulative propagator, consistent with Section 4.1.

Substituting the initial deviation from Eq. (22):

$$\mu^{[\tau]} = P^{[t_{z_j}+1,\tau]} \cdot v^{[t_{z_j}]}, \tag{28}$$

which matches Eq. (11) in Section 4.2.

**Comparison with Sample-Level Unlearning**   The mini-batch formulation differs from sample-level unlearning in the counterfactual model as shown in Table 6.

*Table 6.* Comparison of sample-level and mini-batch unlearning.

|  | Sample-Level | Batch-Level |
| --- | :---: | :---: |
| Counterfactual at $t_{z_j}$ | Replace $z_j$'s contribution with zero-gradient | Skip entire step |
| Initial deviation | $\mu^{[t_{z_j}+1]} = -\frac{\eta_{t_{z_j}}}{|S_{t_{z_j}}|}\nabla_\theta \ell(z_j; \theta^{[t_{z_j}]})$ | $\mu^{[t_{z_j}+1]} = -\eta_{t_{z_j}}\bar{g}^{[t_{z_j}]}$ |
| Storage per step | $O(Bp)$ (per-sample gradients) | $O(p)$ (batch update) |
| Unlearning granularity | Single sample | Entire batch |

The propagation Eq. (25) is identical for both formulations, differing only in the initial deviation magnitude. Batch-level unlearning trades granularity for a $B\times$ reduction in storage, making it practical for large-scale applications.

## C. Detailed Theoretical Analysis

### C.1. Notation and setup

Fix a target mini-batch $S_{t_0}$ used at step $t_0$ in the original run. Let $L_t(\theta)$ denote the (mini-batch) objective used by SGD at step $t$. If weight decay is part of training, we write the regularized objective $\tilde{L}_t(\theta) := L_t(\theta) + \frac{\lambda_{\text{reg}}}{2}\|\theta\|_2^2$ and $\tilde{H}_t(\theta) := \nabla^2 \tilde{L}_t(\theta)$.

- $\theta^{[t]}$: original parameters at step $t$.

- $\theta^{[t]}_{-S_{t_0}}$: counterfactual parameters that skip the update at $t_0$ and then follow the same subsequent mini-batches and learning rates.

- $\mu^{[t]}_{\text{true}} := \theta^{[t]} - \theta^{[t]}_{-S_{t_0}}$: true counterfactual deviation.

- $v^{[t]} := \theta^{[t+1]} - \theta^{[t]} = -\eta_t \nabla \tilde{L}_t(\theta^{[t]})$: the SGD update at step $t$ (for the objective actually used in training).

- $\mu_0 := \mu^{[t_0+1]}_{\text{true}} = \theta^{[t_0+1]} - \theta^{[t_0]} = v^{[t_0]}$.

- $\tilde{H}^{[t]} := \tilde{H}_t(\theta^{[t]})$ and $\tilde{P}^{[t]} := I - \eta_t \tilde{H}^{[t]}$.

- $K^\star := \min\{K, \tau - t_0 - 1\}$ and $t_{\text{end}} := t_0 + K^\star + 1 = \min\{\tau, t_0 + K + 1\}$.

We define the *linearized propagated deviation* $\mu^{[t]}_{\text{lin}}$ by

$$\mu^{[t_0+1]}_{\text{lin}} := \mu_0, \qquad \mu^{[t+1]}_{\text{lin}} := \tilde{P}^{[t]} \mu^{[t]}_{\text{lin}} \ \text{ for } t \geq t_0 + 1, \tag{29}$$

and the truncated (held) deviation used by LMCLEANER is

$$\mu^{[\tau]}_K := \mu^{[t_{\text{end}}]}_{\text{lin}}.$$

### C.2. Assumptions

**Assumption C.1** (Deferred benign re-learning stability and residual)**.** Let $z_j \in S_{t_j}$ be the requested sample and let $C_j := S_{t_j} \setminus \{z_j\}$ denote the benign co-batch samples. Let $\theta^{[\tau]}_{-S_{t_j}}$ be the exact batch-skip counterfactual trajectory and let $\theta^{[\tau]}_{-j}$ be the sample-level counterfactual trajectory that excludes only $z_j$.

Let $\mathsf{C}_j$ denote the deferred benign re-learning operator that reintroduces the benign co-batch samples $C_j$ according to the deferred re-learning schedule. Define

$$\theta^{[\tau]}_{\text{dr},-j} := \mathsf{C}_j(\theta^{[\tau]}_{-S_{t_j}}), \qquad \widehat{\theta}^{[\tau]}_{\text{dr},-j} := \mathsf{C}_j(\widehat{\theta}^{[\tau]}).$$

We assume that, on the local region considered in the analysis, $\mathsf{C}_j$ is $L_{\text{dr}}$-Lipschitz:

$$\|\mathsf{C}_j(\theta) - \mathsf{C}_j(\theta')\|_2 \leq L_{\text{dr}}\|\theta - \theta'\|_2.$$

We further assume that the deferred re-learning residual satisfies

$$\|\mathsf{C}_j(\theta^{[\tau]}_{-S_{t_j}}) - \theta^{[\tau]}_{-j}\|_2 \leq \Gamma_{\text{dr}}(\tau).$$

For certification, we use public upper bounds $\bar{L}_{\text{dr}} \geq L_{\text{dr}}$ and $\bar{\Gamma}_{\text{dr}}(\tau) \geq \Gamma_{\text{dr}}(\tau)$. For subspace-aware release, we additionally assume that the final sample-level mean shift

$$e_{\text{sample}} := \widehat{\theta}^{[\tau]}_{\text{dr},-j} - \theta^{[\tau]}_{-j}$$

satisfies

$$\|(I - \Pi_k)e_{\text{sample}}\|_2 \leq \beta \|e_{\text{sample}}\|_2$$

for public $\Pi_k$ and $\beta$. If this condition is unavailable, we use the isotropic fallback with $\beta = 1$.

**Assumption C.2** (Smoothness / Hessian continuity). For each step $t$, the Hessian of the training objective is $\rho$-Lipschitz:

$$\|\tilde{H}_t(\theta) - \tilde{H}_t(\theta')\|_2 \leq \rho\|\theta - \theta'\|_2, \qquad \forall \theta, \theta'. \tag{30}$$

**Assumption C.3** (Uniform bounded curvature and strict contraction). There exist constants $0 < \nu \leq \tilde{L}$ such that for all steps $t$ and all $\theta$, the Hessian eigenvalues satisfy

$$\nu I \preceq \tilde{H}_t(\theta) \preceq \tilde{L}I. \tag{31}$$

The learning rates satisfy $0 < \eta_t \leq \eta_{\max} < 2/\tilde{L}$. Define

$$\gamma := \max_t \max\{|1 - \eta_t\nu|, |1 - \eta_t\tilde{L}|\} < 1. \tag{32}$$

**Assumption C.4** (Residual subspace concentration). Let $\Pi_k$ be a projector onto a chosen $k$-dimensional *public* subspace (e.g., obtained from a public pilot estimate or fixed public randomness). The tail residual $r^{[\tau]}$ defined in Proposition C.7 satisfies $\|(I - \Pi_k)r^{[\tau]}\|_2 \leq \beta\|r^{[\tau]}\|_2$ for some $\beta \in (0, 1)$.

### C.3. Approximation error bounds

**Lemma C.5** (Second-order Taylor remainder for gradients). *Under Assumption C.2, for any $t$ and any $\theta, \theta'$, letting $\Delta := \theta - \theta'$, we have*

$$\nabla\tilde{L}_t(\theta') = \nabla\tilde{L}_t(\theta) - \tilde{H}_t(\theta)\Delta + r_t, \qquad \|r_t\|_2 \leq \frac{\rho}{2}\|\Delta\|_2^2. \tag{33}$$

*Proof.* Using the integral form of Taylor's theorem,

$$\nabla\tilde{L}_t(\theta - \Delta) = \nabla\tilde{L}_t(\theta) - \left(\int_0^1 \tilde{H}_t(\theta - s\Delta)\, ds\right)\Delta.$$

Add and subtract $\tilde{H}_t(\theta)\Delta$ and define

$$r_t := \left(\int_0^1 \left(\tilde{H}_t(\theta) - \tilde{H}_t(\theta - s\Delta)\right) ds\right)\Delta.$$

By Assumption C.2, $\|\tilde{H}_t(\theta) - \tilde{H}_t(\theta - s\Delta)\|_2 \leq \rho s\|\Delta\|_2$. Therefore,

$$\|r_t\|_2 \leq \int_0^1 \rho s\|\Delta\|_2\, ds \cdot \|\Delta\|_2 = \frac{\rho}{2}\|\Delta\|_2^2.$$

$\square$

**Lemma C.6** (Contraction of the SGD map and deviation decay). *Under Assumption C.3, for each step $t$, the update map $F_t(\theta) := \theta - \eta_t\nabla\tilde{L}_t(\theta)$ is $\gamma$-Lipschitz:*

$$\|F_t(\theta) - F_t(\theta')\|_2 \leq \gamma\|\theta - \theta'\|_2, \qquad \forall \theta, \theta'. \tag{34}$$

*Consequently, for the counterfactual deviation (for all $t \geq t_0 + 1$),*

$$\|\mu_{\mathrm{true}}^{[t]}\|_2 \leq \gamma^{t-(t_0+1)}\|\mu_0\|_2 \leq \|\mu_0\|_2. \tag{35}$$

*Proof.* For any $\theta, \theta'$, define the segment $\theta_s := \theta' + s(\theta - \theta')$. By the fundamental theorem of calculus for vector fields,

$$F_t(\theta) - F_t(\theta') = \int_0^1 \nabla F_t(\theta_s)\, ds\, (\theta - \theta'), \quad \text{where } \nabla F_t(\theta_s) = I - \eta_t\tilde{H}_t(\theta_s).$$

By Assumption C.3, the eigenvalues of $\tilde{H}_t(\theta_s)$ lie in $[\nu, \tilde{L}]$ for all $s$, hence

$$\|\nabla F_t(\theta_s)\|_2 = \|I - \eta_t\tilde{H}_t(\theta_s)\|_2 \leq \max\{|1 - \eta_t\nu|, |1 - \eta_t\tilde{L}|\} \leq \gamma.$$

Therefore,

$$\|F_t(\theta) - F_t(\theta')\|_2 \leq \int_0^1 \|\nabla F_t(\theta_s)\|_2 ds \cdot \|\theta - \theta'\|_2 \leq \gamma \|\theta - \theta'\|_2.$$

Applying this to the two trajectories (which follow the same $F_t$ for all $t \geq t_0 + 1$) yields $\|\mu_{\text{true}}^{[t+1]}\|_2 \leq \gamma \|\mu_{\text{true}}^{[t]}\|_2$ and thus (35) by induction. $\quad\square$

**Proposition C.7** (Tail (zeroth-order hold) residual). *Define the tail residual (the error caused by holding after $t_{\text{end}}$) as*

$$r^{[\tau]} := \mu_{\text{true}}^{[\tau]} - \mu_{\text{true}}^{[t_{\text{end}}]}. \tag{36}$$

*Under Assumption C.3, for any $\tau \geq t_0 + 1$,*

$$\|r^{[\tau]}\|_2 \leq 2\gamma^K \|\mu_0\|_2. \tag{37}$$

*Moreover, if $\tau \leq t_0 + K + 1$ (equivalently $t_{\text{end}} = \tau$), then $r^{[\tau]} = 0$. If Assumption C.4 holds, then $\|(I - \Pi_k)r^{[\tau]}\|_2 \leq \beta\|r^{[\tau]}\|_2$.*

*Proof.* If $\tau \leq t_0 + K + 1$, then $t_{\text{end}} = \tau$ and $r^{[\tau]} = 0$. Otherwise $t_{\text{end}} = t_0 + K + 1$, so by triangle inequality,

$$\|r^{[\tau]}\|_2 \leq \|\mu_{\text{true}}^{[\tau]}\|_2 + \|\mu_{\text{true}}^{[t_{\text{end}}]}\|_2.$$

Applying Lemma C.6 gives $\|\mu_{\text{true}}^{[\tau]}\|_2 \leq \gamma^{\tau - t_0 - 1} \|\mu_0\|_2 \leq \gamma^K \|\mu_0\|_2$ (since $\tau - t_0 - 1 \geq K + 1$) and $\|\mu_{\text{true}}^{[t_{\text{end}}]}\|_2 \leq \gamma^K \|\mu_0\|_2$, hence $\|r^{[\tau]}\|_2 \leq 2\gamma^K \|\mu_0\|_2$. The subspace statement follows directly from Assumption C.4. $\quad\square$

**Lemma C.8** (Trust-region linearization error). *Let $e^{[t]} := \mu_{\text{lin}}^{[t]} - \mu_{\text{true}}^{[t]}$. Under Assumptions C.2–C.3,*

$$\|e^{[t_{\text{end}}]}\|_2 \leq \eta_{\max} \left( \frac{1 - \gamma^{K^\star}}{1 - \gamma} \right) \cdot \frac{\rho}{2} \|\mu_0\|_2^2. \tag{38}$$

*Proof.* For $t \geq t_0 + 1$, the two trajectories both apply the same step-$t$ update:

$$\theta^{[t+1]} = \theta^{[t]} - \eta_t \nabla \tilde{L}_t(\theta^{[t]}), \qquad \theta_{-S_{t_0}}^{[t+1]} = \theta_{-S_{t_0}}^{[t]} - \eta_t \nabla \tilde{L}_t(\theta_{-S_{t_0}}^{[t]}).$$

Thus,

$$\mu_{\text{true}}^{[t+1]} = \mu_{\text{true}}^{[t]} - \eta_t \left( \nabla \tilde{L}_t(\theta^{[t]}) - \nabla \tilde{L}_t(\theta_{-S_{t_0}}^{[t]}) \right).$$

Apply Lemma C.5 with $\theta = \theta^{[t]}$ and $\theta' = \theta_{-S_{t_0}}^{[t]} = \theta^{[t]} - \mu_{\text{true}}^{[t]}$ to obtain

$$\nabla \tilde{L}_t(\theta_{-S_{t_0}}^{[t]}) = \nabla \tilde{L}_t(\theta^{[t]}) - \tilde{H}^{[t]} \mu_{\text{true}}^{[t]} + r_t, \qquad \|r_t\|_2 \leq \frac{\rho}{2} \|\mu_{\text{true}}^{[t]}\|_2^2.$$

Rearranging gives

$$\mu_{\text{true}}^{[t+1]} = (I - \eta_t \tilde{H}^{[t]}) \mu_{\text{true}}^{[t]} + \eta_t r_t = \tilde{P}^{[t]} \mu_{\text{true}}^{[t]} + \eta_t r_t.$$

The linearized dynamics satisfy $\mu_{\text{lin}}^{[t+1]} = \tilde{P}^{[t]} \mu_{\text{lin}}^{[t]}$, so the error recursion is

$$e^{[t+1]} = \tilde{P}^{[t]} e^{[t]} - \eta_t r_t.$$

Taking norms, using $\|\tilde{P}^{[t]}\|_2 \leq \gamma$ and $\eta_t \leq \eta_{\max}$ gives

$$\|e^{[t+1]}\|_2 \leq \gamma \|e^{[t]}\|_2 + \eta_{\max} \frac{\rho}{2} \|\mu_{\text{true}}^{[t]}\|_2^2.$$

By Lemma C.6, $\|\mu_{\text{true}}^{[t]}\|_2 \leq \|\mu_0\|_2$. Also $e^{[t_0+1]} = 0$ since $\mu_{\text{lin}}^{[t_0+1]} = \mu_0 = \mu_{\text{true}}^{[t_0+1]}$. Unrolling for $K^\star$ steps yields (38). $\quad\square$

*Proof of Theorem 5.1.* By definition,

$$\widehat{\theta}^{[\tau]} - \theta^{[\tau]}_{-S_{t_0}} = (\theta^{[\tau]} - \mu_K^{[\tau]}) - (\theta^{[\tau]} - \mu_{\text{true}}^{[\tau]}) = \mu_{\text{true}}^{[\tau]} - \mu_K^{[\tau]}.$$

Since $\mu_K^{[\tau]} = \mu_{\text{lin}}^{[t_{\text{end}}]}$,

$$\mu_{\text{true}}^{[\tau]} - \mu_K^{[\tau]} = \underbrace{\left(\mu_{\text{true}}^{[\tau]} - \mu_{\text{true}}^{[t_{\text{end}}]}\right)}_{r^{[\tau]}} + \underbrace{\left(\mu_{\text{true}}^{[t_{\text{end}}]} - \mu_{\text{lin}}^{[t_{\text{end}}]}\right)}_{-e^{[t_{\text{end}}]}}.$$

Therefore,

$$\|\widehat{\theta}^{[\tau]} - \theta^{[\tau]}_{-S_{t_0}}\|_2 \le \|r^{[\tau]}\|_2 + \|e^{[t_{\text{end}}]}\|_2.$$

Apply Proposition C.7 and Lemma C.8. $\qquad\square$

**Explicit instantiation of $\Delta_{\text{cert}}(K)$.** Combining Proposition C.7 and Lemma C.8, one valid choice is

$$\Delta_{\text{lin}}(K^\star) := \eta_{\max}\left(\frac{1 - \gamma^{K^\star}}{1 - \gamma}\right) \cdot \frac{\rho}{2}\|\mu_0\|_2^2, \qquad R(K) := 2\gamma^K\|\mu_0\|_2,$$

hence $\Delta_{\text{cert}}(K) = \Delta_{\text{lin}}(K^\star) + R(K)$. (If propagation uses an extra $\lambda_{\text{reg}}I$ not present in training, apply Corollary C.9.)

### C.4. Stability-only regularization

This subsection justifies the extra additive term stated in the main text remark.

**Corollary C.9** (Extra bias when $\lambda_{\text{reg}}$ is used only in propagation)**.** *Assume the training dynamics use the unregularized objective $L_t$ but the propagation uses $\tilde{H}^{[t]} = H^{[t]} + \lambda_{\text{reg}}I$. Assume further that $0 < \eta_t \le \eta_{\max} < 2/(L + \lambda_{\text{reg}})$ so that*

$$\tilde{\gamma} := \max_t \max\{|1 - \eta_t(\nu + \lambda_{\text{reg}})|, \ |1 - \eta_t(L + \lambda_{\text{reg}})|\} < 1,$$

*and thus $\|\tilde{P}^{[t]}\|_2 \le \tilde{\gamma}$. Under Assumptions C.2–C.3 (interpreted for $H$ in the true dynamics), the bound in Lemma C.8 becomes*

$$\|e^{[t_{\text{end}}]}\|_2 \le \eta_{\max}\left(\frac{1 - \tilde{\gamma}^{K^\star}}{1 - \tilde{\gamma}}\right)\left(\lambda_{\text{reg}}\|\mu_0\|_2 + \frac{\rho}{2}\|\mu_0\|_2^2\right),$$

*hence Theorem 5.1 gains the extra additive term proportional to $\lambda_{\text{reg}}\|\mu_0\|_2$.*

*Proof.* The derivation follows Lemma C.8, except that the true deviation satisfies $\mu_{\text{true}}^{[t+1]} = (I - \eta_t H^{[t]})\mu_{\text{true}}^{[t]} + \eta_t r_t$, while the propagation uses $\tilde{P}^{[t]} = I - \eta_t(H^{[t]} + \lambda_{\text{reg}}I)$. Thus,

$$e^{[t+1]} = \tilde{P}^{[t]}e^{[t]} - \eta_t\lambda_{\text{reg}}\mu_{\text{true}}^{[t]} - \eta_t r_t.$$

Taking norms, using $\|\tilde{P}^{[t]}\|_2 \le \tilde{\gamma}$, $\eta_t \le \eta_{\max}$ and $\|\mu_{\text{true}}^{[t]}\|_2 \le \|\mu_0\|_2$ gives the recursion

$$\|e^{[t+1]}\|_2 \le \tilde{\gamma}\|e^{[t]}\|_2 + \eta_{\max}\left(\lambda_{\text{reg}}\|\mu_0\|_2 + \frac{\rho}{2}\|\mu_0\|_2^2\right).$$

Unrolling with $e^{[t_0+1]} = 0$ yields the claim. $\qquad\square$

### C.5. Certified Privacy Guarantee

We certify indistinguishability for the *released* parameters after applying a public, data-independent noise mechanism. Define the public post-processing mechanism

$$\mathcal{M}(\theta) := \theta + \xi, \qquad \xi \sim \mathcal{N}(0, \Sigma),$$

where $\Sigma = \sigma_\parallel^2\Pi_k + \sigma_\perp^2(I - \Pi_k)$ is fixed by public parameters. Accordingly, the reference retraining distribution is $\mathcal{M}(\theta^{[\tau]}_{-S_{t_0}})$. This matches Definition 3.1 when the released retraining procedure is $\mathcal{R}_{\mathcal{M}}(D') := \mathcal{M}(\mathcal{R}(D'))$.

**Public calibration via update-norm clipping.** Let $v^{[t]} := \theta^{[t+1]} - \theta^{[t]}$ be the (mini-batch) SGD update. With global gradient/update-norm clipping at threshold $C$ (a public hyperparameter), we have $\|v^{[t]}\|_2 \leq \eta_t C$, hence

$$\|\mu_0\|_2 = \|v^{[t_0]}\|_2 \leq \eta_{t_0} C \leq \eta_{\max} C =: G_{\max}. \tag{39}$$

We define $\bar{\Delta}_{\mathrm{cert}}(K)$ by substituting $\|\mu_0\|_2 \leftarrow G_{\max}$ in the explicit bound of $\Delta_{\mathrm{cert}}(K)$ (see Appendix C.3), making the noise scale data-independent.

**Assumption C.10** (Public mechanism parameters)**.** The projector $\Pi_k \in \mathbb{R}^{p \times p}$ is fixed in advance (or depends only on public randomness), and is independent of the dataset and the removal request. Moreover, the noise scales $(\sigma_\|, \sigma_\perp)$ are calibrated using a *public* (data-independent) bound $\bar{\Delta}_{\mathrm{cert}}(K)$. Equivalently, the noise law $\xi \sim \mathcal{N}(0, \Sigma)$ with $\Sigma = \sigma_\|^2 \Pi_k + \sigma_\perp^2 (I - \Pi_k)$ is identical when comparing the unlearning output and the retraining reference.

**Assumption C.11** (Public calibration bound)**.** There exists a public constant $\bar{\Delta}_{\mathrm{cert}}(K)$ such that for all datasets and requests, the deterministic mean shift satisfies

$$\|e\|_2 = \|\widehat{\theta}^{[\tau]} - \theta^{[\tau]}_{-S_{t_0}}\|_2 \leq \Delta_{\mathrm{cert}}(K) \leq \bar{\Delta}_{\mathrm{cert}}(K).$$

(One can instantiate $\bar{\Delta}_{\mathrm{cert}}(K)$ by upper bounding $\|\mu_0\|_2$ using, e.g., a known update/gradient-norm bound from clipping.)

**Assumption C.12** (Error concentration in the top-$k$ subspace)**.** Let $e := \widehat{\theta}^{[\tau]} - \theta^{[\tau]}_{-S_{t_0}}$ be the deterministic mean shift between LMCLEANER (before noise) and exact counterfactual retraining. There exists a public $\beta \in (0, 1]$ such that

$$\|(I - \Pi_k)e\|_2 \leq \beta \|e\|_2.$$

## C.6. Proof of Certified Unlearning (Theorem 5.2)

*Proof.* We prove $(\varepsilon, \delta)$-certified unlearning by bounding the deterministic mean shift and applying the Gaussian mechanism on two orthogonal blocks.

**Step 1: Bounded mean shift.** Define

$$m_{\mathcal{A}} := \widehat{\theta}^{[\tau]}, \qquad m_{\mathcal{R}} := \theta^{[\tau]}_{-S_{t_0}}, \qquad e := m_{\mathcal{A}} - m_{\mathcal{R}}.$$

By Theorem 5.1, $\|e\|_2 \leq \Delta_{\mathrm{cert}}(K)$, hence by Assumption C.11, $\|e\|_2 \leq \bar{\Delta}_{\mathrm{cert}}(K)$.

**Step 2: Orthonormal coordinates.** Let $U \in \mathbb{R}^{p \times k}$ have orthonormal columns spanning $\mathrm{Im}(\Pi_k)$ so that $\Pi_k = UU^\top$. Let $V \in \mathbb{R}^{p \times (p-k)}$ span the orthogonal complement so that $I - \Pi_k = VV^\top$, and $U^\top V = 0$.

Consider the block mechanism with independent noises:

$$\mathcal{M}(m) := \left( U^\top m + \mathcal{N}(0, \sigma_\|^2 I_k), \ V^\top m + \mathcal{N}(0, \sigma_\perp^2 I_{p-k}) \right).$$

Its reconstruction $Uy_\| + Vy_\perp$ is distributed as $m + \xi$ where $\xi \sim \mathcal{N}(0, \sigma_\|^2 \Pi_k + \sigma_\perp^2 (I - \Pi_k))$. Thus, it suffices to prove $(\varepsilon, \delta)$-DP for $\mathcal{M}$; the final output inherits it by post-processing.

**Step 3: Sensitivity in each block.** For two inputs $m, m'$ with difference $e = m - m'$,

$$\Delta_\| := \|U^\top e\|_2 = \|\Pi_k e\|_2 \leq \|e\|_2 \leq \bar{\Delta}_{\mathrm{cert}}(K).$$

By Assumption C.12,

$$\Delta_\perp := \|V^\top e\|_2 = \|(I - \Pi_k)e\|_2 \leq \beta \|e\|_2 \leq \beta \bar{\Delta}_{\mathrm{cert}}(K).$$

**Step 4: Gaussian mechanism on each block with budget splitting.** We use the classical Gaussian mechanism bound, which applies for $\varepsilon_b \in (0, 1)$ and $\delta_b \in (0, 1)$. Since we allocate $(\varepsilon_b, \delta_b) = (\varepsilon/2, \delta/2)$ to each orthogonal block, this proof assumes $0 < \varepsilon < 2$ and $0 < \delta < 1$. For larger $\varepsilon$, the same argument can be used with the analytic Gaussian mechanism by replacing the closed-form constants below with the corresponding analytic calibration.

For the top-$k$ block, $\Delta_\| \le \bar{\Delta}_{\mathrm{cert}}(K)$. Thus, setting

$$\sigma_\| = \frac{2\bar{\Delta}_{\mathrm{cert}}(K)}{\varepsilon}\sqrt{2\log\left(\frac{2.5}{\delta}\right)}$$

ensures $(\varepsilon/2, \delta/2)$-DP for this block.

For the orthogonal-complement block, $\Delta_\perp \le \beta\bar{\Delta}_{\mathrm{cert}}(K)$. Thus, setting

$$\sigma_\perp = \frac{2\beta\bar{\Delta}_{\mathrm{cert}}(K)}{\varepsilon}\sqrt{2\log\left(\frac{2.5}{\delta}\right)}$$

ensures $(\varepsilon/2, \delta/2)$-DP for this block.

**Step 5: Composition and conclusion.** By basic composition, $\mathcal{M}$ is $(\varepsilon, \delta)$-DP. Therefore, for all measurable $W$,

$$\Pr[m_\mathcal{A} + \xi \in W] \le e^\varepsilon \Pr[m_\mathcal{R} + \xi \in W] + \delta, \quad \Pr[m_\mathcal{R} + \xi \in W] \le e^\varepsilon \Pr[m_\mathcal{A} + \xi \in W] + \delta,$$

which proves that $\mathcal{M}(m_\mathcal{A})$ and $\mathcal{M}(m_\mathcal{R})$ are $(\varepsilon, \delta)$-indistinguishable, i.e., the released output is $(\varepsilon, \delta)$-certified with respect to the noisy retraining reference $\mathcal{M}(\theta^{[\tau]}_{-S_{t_0}})$. Assumption C.10 ensures the mechanism is identical on both sides.

**Fallback: isotropic noise.** If Assumption C.12 is unavailable, set $\Pi_k = I$ (or $\beta = 1$) and use isotropic noise $\xi \sim \mathcal{N}(0, \sigma^2 I_p)$ with $\sigma \ge \frac{\bar{\Delta}_{\mathrm{cert}}(K)}{\varepsilon}\sqrt{2\log(1.25/\delta)}$, which still yields $(\varepsilon, \delta)$-certified unlearning. $\qquad\square$

### C.6.1. TRADE-OFF ANALYSIS

**Proposition C.13** (Approximate optimal window selection)**.** *Let $\bar{G} := G_{\max}$ be the public bound on $\|\mu_0\|_2$ (Eq. (39)) and define*

$$\bar{\phi} := \eta_{\max}\left(\lambda_{\mathrm{reg}}\bar{G} + \frac{\rho}{2}\bar{G}^2\right).$$

*For integer $K \in \mathbb{Z}_{\ge 0}$, using $\sum_{i=0}^{K-1}\gamma^i \le K$ yields the public upper bound*

$$\bar{\Delta}^{\mathrm{ub}}_{\mathrm{cert}}(K) := K\bar{\phi} + 2\bar{G}\gamma^K.$$

*Treating $K$ as a continuous variable gives a stationary point if $2\bar{G}|\ln\gamma| > \bar{\phi}$, namely*

$$K^\star_{\mathrm{cont}} = \frac{1}{|\ln\gamma|}\ln\left(\frac{2\bar{G}|\ln\gamma|}{\bar{\phi}}\right),$$

*otherwise the minimizer is $K^\star_{\mathrm{cont}} = 0$. For integer windows, take $K^\star = \max\{0, \mathrm{round}(K^\star_{\mathrm{cont}})\}$.*

*Proof.* Differentiate $\bar{\Delta}^{\mathrm{ub}}_{\mathrm{cert}}(K) = K\bar{\phi} + 2\bar{G}e^{K\ln\gamma}$:

$$\frac{\partial\bar{\Delta}^{\mathrm{ub}}_{\mathrm{cert}}}{\partial K} = \bar{\phi} + 2\bar{G}\gamma^K\ln\gamma.$$

Since $\ln\gamma < 0$, a stationary point exists iff $2\bar{G}|\ln\gamma| > \bar{\phi}$, yielding $\gamma^{K^\star_{\mathrm{cont}}} = \bar{\phi}/(2\bar{G}|\ln\gamma|)$ and the stated expression. $\qquad\square$

**Corollary C.14** (Sample-level certified unlearning with deferred re-learning)**.** *Let $z_j \in S_{t_j}$ be the requested sample and let $\theta^{[\tau]}_{-j}$ denote the sample-level counterfactual trajectory that excludes only $z_j$. Suppose Theorem 5.1 and Assumption C.1 hold. Then the compensated deterministic output*

$$\widehat{\theta}^{[\tau]}_{\mathrm{dr},-j} := \mathsf{C}_j(\widehat{\theta}^{[\tau]})$$

*satisfies*

$$\|\widehat{\theta}^{[\tau]}_{\mathrm{dr},-j} - \theta^{[\tau]}_{-j}\|_2 \le \|\mathsf{C}_j(\widehat{\theta}^{[\tau]}) - \mathsf{C}_j(\theta^{[\tau]}_{-S_{t_j}})\|_2 + \|\mathsf{C}_j(\theta^{[\tau]}_{-S_{t_j}}) - \theta^{[\tau]}_{-j}\|_2$$
$$\le L_{\mathrm{dr}}\Delta_{\mathrm{cert}}(K) + \Gamma_{\mathrm{dr}}(\tau).$$

*Define the public sample-level sensitivity bound*

$$\bar{\Delta}_{\mathrm{sample}}(K) := \bar{L}_{\mathrm{dr}}\bar{\Delta}_{\mathrm{cert}}(K) + \bar{\Gamma}_{\mathrm{dr}}(\tau).$$

*If the Gaussian release noise in Eq.* (14) *is calibrated using* $\bar{\Delta}_{\mathrm{sample}}(K)$ *instead of* $\bar{\Delta}_{\mathrm{cert}}(K)$*, then*

$$\mathcal{M}(\widehat{\theta}^{[\tau]}_{\mathrm{dr},-j})$$

*is* $(\varepsilon, \delta)$*-indistinguishable from the noisy sample-level reference*

$$\mathcal{M}(\theta^{[\tau]}_{-j}).$$

*Therefore,* LMCLEANER *achieves sample-level* $(\varepsilon, \delta)$*-certified unlearning for the compensated release.*

# D. Multi-Epoch Extension

The main text first analyzes the single-pass setting, where each training sample is used at most once before the unlearning request. We now extend the analysis to multi-epoch fine-tuning, where the same target sample may appear multiple times. The purpose of this appendix is to show that the proposed influence-propagation correction remains valid when a target sample contributes to multiple mini-batch updates across epochs.

The analysis is conducted at the mini-batch atomic level. That is, whenever the target sample $z_j$ appears in a mini-batch, the affected mini-batch update is treated as one influence impulse. The connection between this batch-level counterfactual and sample-level deletion is made through deferred benign-sample compensation, formalized in Assumption D.2.

Let

$$\mathcal{T}_j := \{t_1 < t_2 < \cdots < t_M\}$$

be the set of training steps before the unlearning request at which a mini-batch containing $z_j$ is used. The unlearning request arrives at step $\tau \geq t_M + 1$. Let $S_{t_m}$ denote the mini-batch containing $z_j$ at occurrence $t_m$; these mini-batches may differ across epochs.

Throughout this appendix, we use the same smoothness, gradient clipping, and local contraction assumptions as in the main theoretical analysis. In particular, gradient clipping gives

$$\|v^{[t]}\|_2 \leq G_{\max} := \eta_{\max} C,$$

where

$$v^{[t]} := \theta^{[t+1]} - \theta^{[t]} = -\eta_t \bar{g}^{[t]}.$$

**Regularization convention.** In the multi-epoch extension, $\lambda_{\mathrm{reg}}$ denotes a *stability-only* regularization term used by the influence propagation operator. Thus the true training dynamics are written with $L_t$, whereas the propagated linear operator uses $\tilde{H}^{[t]} = H^{[t]} + \lambda_{\mathrm{reg}} I$. If weight decay is part of the actual training objective, it should be absorbed into $L_t$ and $H^{[t]}$; in that case, set $\lambda_{\mathrm{reg}} = 0$ in the extra-bias terms below.

## D.1. Batch-Skip and Sample-Deletion Counterfactuals

**Definition D.1** (Multi-epoch batch-skip counterfactual). The *multi-epoch batch-skip counterfactual trajectory* $\{\theta_*^{[t]}\}_{t=0}^{\tau}$ is defined by skipping the entire mini-batch update at every occurrence step $t_m \in \mathcal{T}_j$:

$$\theta_*^{[t+1]} = \begin{cases} \theta_*^{[t]}, & t \in \mathcal{T}_j, \\ \theta_*^{[t]} - \eta_t \nabla L_t(\theta_*^{[t]}), & t \notin \mathcal{T}_j, \end{cases} \tag{40}$$

where

$$L_t(\theta) := \frac{1}{|S_t|} \sum_{i \in S_t} \ell(z_i; \theta).$$

Before the first occurrence, the original and counterfactual trajectories coincide:

$$\theta_*^{[t]} = \theta^{[t]}, \qquad t \leq t_1.$$

The trajectory $\theta_*^{[\tau]}$ corresponds to removing all mini-batch updates that contain $z_j$. The ideal sample-deletion trajectory, denoted by $\theta_{-j}^{[\tau]}$, excludes only $z_j$ while preserving the contribution of benign samples that appeared in the same mini-batches. Since LMCleaner operates on mini-batch updates, we connect these two trajectories through deferred re-learning.

**Assumption D.2** (Deferred benign-sample compensation with public bounds). Let $\theta_{-j}^{[\tau]}$ denote the ideal sample-deletion trajectory in which $z_j$ is excluded at every occurrence, while all benign samples in $S_{t_m} \setminus \{z_j\}$ remain in the training stream. Let $\theta_*^{[\tau]}$ denote the multi-epoch batch-skip counterfactual in Definition D.1.

There exists a deferred re-learning operator C, applied to the benign samples

$$\{S_{t_m} \setminus \{z_j\}\}_{m=1}^{M},$$

such that

$$\|C(\theta_*^{[\tau]}) - \theta_{-j}^{[\tau]}\|_2 \leq \Delta_{\text{comp}}. \tag{41}$$

Moreover, $C$ is $L_{\text{comp}}$-Lipschitz:

$$\|C(\theta) - C(\theta')\|_2 \leq L_{\text{comp}}\|\theta - \theta'\|_2. \tag{42}$$

For certification, we use public data-independent upper bounds $\bar{L}_{\text{comp}} \geq L_{\text{comp}}$ and $\bar{\Delta}_{\text{comp}} \geq \Delta_{\text{comp}}$. In the ideal compensation regime, $\bar{\Delta}_{\text{comp}} = 0$ and $\bar{L}_{\text{comp}} = 1$.

Assumption D.2 isolates the only conceptual gap between mini-batch atomic removal and sample-level deletion. The remainder of this appendix proves that the multi-epoch batch-level correction is accurate; Corollary D.10 then transfers this guarantee to the sample-level trajectory under deferred compensation.

## D.2. Multi-Epoch Deviation Recursion

Define the true batch-skip deviation

$$\mu_*^{[t]} := \theta^{[t]} - \theta_*^{[t]}.$$

At an occurrence step $t \in \mathcal{T}_j$, the original trajectory applies the mini-batch update while the counterfactual skips it. Hence

$$\mu_*^{[t+1]} = \mu_*^{[t]} + v^{[t]}, \qquad t \in \mathcal{T}_j. \tag{43}$$

At a non-occurrence step $t \notin \mathcal{T}_j$, both trajectories use the same mini-batch loss $L_t$, giving

$$\mu_*^{[t+1]} = \mu_*^{[t]} - \eta_t \left( \nabla L_t(\theta^{[t]}) - \nabla L_t(\theta_*^{[t]}) \right). \tag{44}$$

Let

$$H^{[t]} := \nabla^2 L_t(\theta^{[t]}), \qquad \tilde{H}^{[t]} := H^{[t]} + \lambda_{\text{reg}} I,$$

and

$$\tilde{P}^{[t]} := I - \eta_t \tilde{H}^{[t]}.$$

For non-occurrence steps, Taylor expansion around $\theta^{[t]}$ gives

$$\nabla L_t(\theta^{[t]}) - \nabla L_t(\theta_*^{[t]}) = H^{[t]} \mu_*^{[t]} + q_t,$$

with

$$\|q_t\|_2 \leq \frac{\rho}{2} \|\mu_*^{[t]}\|_2^2$$

under Hessian-Lipschitz smoothness. Since the algorithm propagates with the regularized Hessian $\tilde{H}^{[t]}$, the true recursion can be written as

$$\mu_*^{[t+1]} = \tilde{P}^{[t]} \mu_*^{[t]} + \eta_t r_t^{(\text{multi})}, \qquad t \notin \mathcal{T}_j, \tag{45}$$

where the effective remainder satisfies

$$\|r_t^{(\text{multi})}\|_2 \leq \lambda_{\text{reg}} \|\mu_*^{[t]}\|_2 + \frac{\rho}{2} \|\mu_*^{[t]}\|_2^2. \tag{46}$$

When $\lambda_{\text{reg}} = 0$, this reduces to the standard second-order Taylor remainder.

## D.3. Masked Linearized Decomposition

To handle later occurrences correctly, define the masked step operator

$$A_t := \begin{cases} I, & t \in \mathcal{T}_j, \\ \tilde{P}^{[t]}, & t \notin \mathcal{T}_j, \end{cases} \tag{47}$$

and the masked cumulative propagator

$$\Phi_{\mathcal{T}_j}^{[a,b]} := A_{b-1} A_{b-2} \cdots A_a, \qquad a < b. \tag{48}$$

The masking is important: if another occurrence of $z_j$ appears inside the propagation window, that step is not propagated through an HVP; it is treated as an identity step, because its effect is represented by a separate additive impulse.

The linearized multi-epoch deviation is defined by dropping the remainder term in Eq. (45):

$$\mu_{\text{lin}}^{[t+1]} = A_t \mu_{\text{lin}}^{[t]} + \mathbf{1}[t \in \mathcal{T}_j] v^{[t]}, \qquad \mu_{\text{lin}}^{[t]} = 0 \text{ for } t \leq t_1. \tag{49}$$

**Proposition D.3** (Multi-occurrence additive influence decomposition). *The linearized multi-epoch deviation decomposes exactly as*

$$\mu_{\text{lin}}^{[\tau]} = \sum_{m=1}^{M} \Phi_{\mathcal{T}_j}^{[t_m+1,\tau]} v^{[t_m]}. \tag{50}$$

*Proof.* Eq. (49) is a linear recursion with additive impulses at occurrence steps. Unrolling the recursion from $t_1$ to $\tau$ gives one propagated impulse for each $t_m \in \mathcal{T}_j$. The propagation after $t_m$ is exactly $\Phi_{\mathcal{T}_j}^{[t_m+1,\tau]}$, because later occurrence steps are masked as identity operators. $\square$

When $M = 1$, Proposition D.3 reduces to the single-pass influence propagation formula in the main text.

### D.4. Multi-Epoch LMCleaner Approximation

For each occurrence $t_m$, define

$$K_m^\star := \min\{K, \tau - t_m - 1\}, \qquad t_{\text{end},m} := t_m + K_m^\star + 1. \tag{51}$$

The multi-epoch LMCleaner correction is obtained by truncating each occurrence separately and summing the resulting corrections:

$$\mu_{\text{agg}}^{[\tau]} := \sum_{m=1}^{M} \Phi_{\mathcal{T}_j}^{[t_m+1,t_{\text{end},m}]} v^{[t_m]}, \qquad \widehat{\theta}^{[\tau]} := \theta^{[\tau]} - \mu_{\text{agg}}^{[\tau]}. \tag{52}$$

Operationally, for each occurrence $t_m$, the algorithm initializes the correction with $v^{[t_m]}$ and propagates it for at most $K$ steps. If a later occurrence step $t_r \in \{t_{m+1}, \ldots, t_M\}$ is encountered inside this window, the correction is left unchanged at that step. Otherwise, the correction is propagated using the regularized HVP update

$$v \leftarrow v - \eta_s(\tilde{H}^{[s]} v).$$

The final multi-epoch correction is the sum of all per-occurrence corrections.

### D.5. Multi-Epoch Approximation Error

The approximation error decomposes as

$$\widehat{\theta}^{[\tau]} - \theta_*^{[\tau]} = \left(\mu_*^{[\tau]} - \mu_{\text{lin}}^{[\tau]}\right) + \left(\mu_{\text{lin}}^{[\tau]} - \mu_{\text{agg}}^{[\tau]}\right). \tag{53}$$

The first term is the multi-epoch linearization error. The second term is the multi-epoch truncation error.

We use the following local stability condition for the true nonlinear dynamics.

**Assumption D.4** (Local non-expansiveness of non-occurrence updates). For every non-occurrence step $t \notin \mathcal{T}_j$, the true SGD update map

$$F_t(\theta) := \theta - \eta_t \nabla L_t(\theta)$$

is non-expansive on the local region visited by the original and counterfactual trajectories:

$$\|F_t(\theta) - F_t(\theta')\|_2 \leq \|\theta - \theta'\|_2. \tag{54}$$

In addition, the regularized linear propagator satisfies

$$\|\tilde{P}^{[t]}\|_2 \leq \gamma, \qquad 0 < \gamma < 1, \tag{55}$$

for all non-occurrence steps in the local analysis region.

**Lemma D.5** (Deviation bound under local non-expansiveness). *Under gradient clipping and Assumption D.4,*

$$\sup_{t \le \tau} \|\mu_*^{[t]}\|_2 \le M G_{\max}. \tag{56}$$

*Proof.* Before the first occurrence, $\mu_*^{[t]} = 0$. At each occurrence step $t_m \in \mathcal{T}_j$, Eq. (43) gives

$$\|\mu_*^{[t_m+1]}\|_2 \le \|\mu_*^{[t_m]}\|_2 + \|v^{[t_m]}\|_2 \le \|\mu_*^{[t_m]}\|_2 + G_{\max}.$$

At each non-occurrence step, Assumption D.4 implies

$$\|\mu_*^{[t+1]}\|_2 = \|F_t(\theta^{[t]}) - F_t(\theta_*^{[t]})\|_2 \le \|\mu_*^{[t]}\|_2.$$

Thus the deviation can increase only at occurrence steps, and each such increase is at most $G_{\max}$. There are $M$ occurrences, so the result follows. □

**Lemma D.6** (Multi-epoch linearization error). *Let $U_M := M G_{\max}$. Under Hessian-Lipschitz smoothness, Assumption D.4, and gradient clipping,*

$$\|\mu_{\mathrm{lin}}^{[\tau]} - \mu_*^{[\tau]}\|_2 \le \Delta_{\mathrm{lin}}^{(\mathrm{multi})}(M), \tag{57}$$

*where*

$$\Delta_{\mathrm{lin}}^{(\mathrm{multi})}(M) := \frac{\eta_{\max}}{1 - \gamma} \left( \lambda_{\mathrm{reg}} U_M + \frac{\rho}{2} U_M^2 \right). \tag{58}$$

*In the unregularized case $\lambda_{\mathrm{reg}} = 0$, this becomes*

$$\Delta_{\mathrm{lin}}^{(\mathrm{multi})}(M) = \frac{\eta_{\max} \rho}{2(1 - \gamma)} (M G_{\max})^2.$$

*Proof.* Let

$$e^{[t]} := \mu_{\mathrm{lin}}^{[t]} - \mu_*^{[t]}.$$

At occurrence steps $t \in \mathcal{T}_j$, both the true deviation and the linearized deviation receive the same additive impulse $v^{[t]}$, so

$$e^{[t+1]} = e^{[t]}.$$

At non-occurrence steps, Eqs. (45) and (49) give

$$e^{[t+1]} = \tilde{P}^{[t]} e^{[t]} - \eta_t r_t^{(\mathrm{multi})}.$$

Taking norms and applying Assumption D.4 yields

$$\|e^{[t+1]}\|_2 \le \gamma \|e^{[t]}\|_2 + \eta_{\max} \|r_t^{(\mathrm{multi})}\|_2.$$

Using Eq. (46) and Lemma D.5,

$$\|r_t^{(\mathrm{multi})}\|_2 \le \lambda_{\mathrm{reg}} U_M + \frac{\rho}{2} U_M^2.$$

Unrolling the geometric recursion gives

$$\|e^{[\tau]}\|_2 \le \frac{\eta_{\max}}{1 - \gamma} \left( \lambda_{\mathrm{reg}} U_M + \frac{\rho}{2} U_M^2 \right),$$

which proves the claim. □

We now bound the truncation error. For each occurrence $t_m$, define the set of non-occurrence propagation steps inside its truncation window:

$$\nu_m(K) := \sum_{s=t_m+1}^{t_{\mathrm{end},m}-1} \mathbf{1}[s \notin \mathcal{T}_j]. \tag{59}$$

Thus $\nu_m(K)$ counts how many genuinely contractive HVP propagation steps are performed for the $m$-th occurrence. Occurrence steps inside the window are masked as identity and therefore do not contribute to this count.

**Lemma D.7** (Multi-epoch truncation error). *Define the set of actually truncated occurrences*

$$\mathcal{I}_K := \{m \in \{1, \dots, M\} : t_m + K + 1 < \tau\}.$$

*Then*

$$\|\mu_{\mathrm{lin}}^{[\tau]} - \mu_{\mathrm{agg}}^{[\tau]}\|_2 \le \Delta_{\mathrm{trunc}}^{(\mathrm{multi})}(K, M), \tag{60}$$

*where*

$$\Delta_{\mathrm{trunc}}^{(\mathrm{multi})}(K, M) := 2G_{\max} \sum_{m \in \mathcal{I}_K} \gamma^{\nu_m(K)}. \tag{61}$$

*Moreover, since*

$$\nu_m(K) \ge \max\{0, K - (M - m)\},$$

*we have the coarse public bound*

$$\Delta_{\mathrm{trunc}}^{(\mathrm{multi})}(K, M) \le 2M\gamma^{\max\{0, K - M + 1\}} G_{\max}. \tag{62}$$

*Proof.* For $m \notin \mathcal{I}_K$, $t_{\mathrm{end},m} = \tau$, so the $m$-th occurrence contributes no truncation error. Consider $m \in \mathcal{I}_K$. The full linearized contribution is

$$\Phi_{\mathcal{T}_j}^{[t_m+1,\tau]} v^{[t_m]},$$

whereas LMCleaner uses the truncated contribution

$$\Phi_{\mathcal{T}_j}^{[t_m+1,t_{\mathrm{end},m}]} v^{[t_m]}.$$

Let

$$x_m := \Phi_{\mathcal{T}_j}^{[t_m+1,t_{\mathrm{end},m}]} v^{[t_m]}.$$

By construction, exactly $\nu_m(K)$ operators inside this window are contractive propagators with norm at most $\gamma$, and the remaining masked operators are identities. Hence

$$\|x_m\|_2 \le \gamma^{\nu_m(K)} G_{\max}.$$

The post-window masked propagator has operator norm at most 1, so

$$\left\| \Phi_{\mathcal{T}_j}^{[t_m+1,\tau]} v^{[t_m]} - \Phi_{\mathcal{T}_j}^{[t_m+1,t_{\mathrm{end},m}]} v^{[t_m]} \right\|_2 \le 2\|x_m\|_2 \le 2\gamma^{\nu_m(K)} G_{\max}.$$

Summing over all $m \in \mathcal{I}_K$ proves Eq. (61).

For the coarse bound, within the $K$ steps after $t_m$, at most $M - m$ steps can be later occurrences. Therefore at least $\max\{0, K - (M - m)\}$ steps are non-occurrence steps, which gives

$$\nu_m(K) \ge \max\{0, K - (M - m)\}.$$

Since $\gamma \in (0, 1)$,

$$\gamma^{\nu_m(K)} \le \gamma^{\max\{0, K - (M - m)\}} \le \gamma^{\max\{0, K - M + 1\}}.$$

Summing over at most $M$ terms yields Eq. (62). $\qquad\square$

**Theorem D.8** (Multi-epoch batch-level approximation error). *Let $U_M := MG_{\max}$. Under gradient clipping, Hessian-Lipschitz smoothness, and Assumption D.4, the multi-epoch LMCleaner estimator satisfies*

$$\|\widehat{\theta}^{[\tau]} - \theta_*^{[\tau]}\|_2 \le \Delta_{\mathrm{ME}}(K, M), \tag{63}$$

*where*

$$\Delta_{\mathrm{ME}}(K, M) := \frac{\eta_{\max}}{1 - \gamma} \left( \lambda_{\mathrm{reg}} U_M + \frac{\rho}{2} U_M^2 \right) + 2G_{\max} \sum_{m \in \mathcal{I}_K} \gamma^{\nu_m(K)}. \tag{64}$$

*A public worst-case upper bound is*

$$\bar{\Delta}_{\mathrm{ME}}(K, M) := \frac{\eta_{\max}}{1 - \gamma} \left( \lambda_{\mathrm{reg}} MG_{\max} + \frac{\rho}{2} (MG_{\max})^2 \right) + 2M\gamma^{\max\{0, K - M + 1\}} G_{\max}. \tag{65}$$

*Proof.* Combine the decomposition in Eq. (53) with Lemmas D.6 and D.7. $\qquad\square$

## D.6. Large-Gap Regime

The worst-case truncation bound in Eq. (62) is conservative. In standard multi-epoch fine-tuning, consecutive occurrences of the same sample are typically separated by a large number of training steps. When the truncation window is short, this yields a much sharper bound.

**Corollary D.9** (Large-gap multi-epoch regime). *Assume that no later occurrence appears inside the $K$-step truncation window of any earlier occurrence, i.e.,*

$$t_{m+1} - t_m > K, \qquad m = 1, \ldots, M - 1.$$

*Then, for every truncated occurrence $m \in \mathcal{I}_K$,*

$$\nu_m(K) = K,$$

*and therefore*

$$\Delta_{\text{trunc}}^{(\text{multi})}(K, M) \leq 2M\gamma^K G_{\max}. \tag{66}$$

*Consequently,*

$$\Delta_{\text{ME}}(K, M) \leq \frac{\eta_{\max}}{1 - \gamma} \left( \lambda_{\text{reg}} M G_{\max} + \frac{\rho}{2} (M G_{\max})^2 \right) + 2M\gamma^K G_{\max}. \tag{67}$$

*Proof.* If $t_{m+1} - t_m > K$, then the set $\{t_m + 1, \ldots, t_m + K\}$ contains no later occurrence step. Therefore all $K$ steps in the truncation window are non-occurrence propagation steps, and $\nu_m(K) = K$. Substituting this into Eq. (61) gives Eq. (66). The full bound follows from Theorem D.8. $\square$

This corollary explains why a small $K$ can remain effective across multiple epochs. The truncation residual still decays exponentially in $K$, with only a linear factor in the number of occurrences $M$. The quadratic term in Eq. (67) is a worst-case linearization bound; in practice, the deviation often contracts substantially between epochs before the next occurrence.

## D.7. Sample-Level Guarantee under Deferred Compensation

We now transfer the batch-level result to the sample-level trajectory under Assumption D.2. Define the compensated LMCleaner output as

$$\widehat{\theta}_{\text{comp}}^{[\tau]} := \mathsf{C}(\widehat{\theta}^{[\tau]}).$$

**Corollary D.10** (Multi-epoch sample-level approximation under deferred compensation). *Under Assumption D.2 and Theorem D.8,*

$$\|\widehat{\theta}_{\text{comp}}^{[\tau]} - \theta_{-j}^{[\tau]}\|_2 \leq L_{\text{comp}} \Delta_{\text{ME}}(K, M) + \Delta_{\text{comp}}. \tag{68}$$

*In the ideal compensation regime $L_{\text{comp}} = 1$ and $\Delta_{\text{comp}} = 0$, the multi-epoch batch-level correction is equivalent to sample-level deletion up to $\Delta_{\text{ME}}(K, M)$.*

*Proof.* By the triangle inequality,

$$\|\mathsf{C}(\widehat{\theta}^{[\tau]}) - \theta_{-j}^{[\tau]}\|_2 \leq \|\mathsf{C}(\widehat{\theta}^{[\tau]}) - \mathsf{C}(\theta_*^{[\tau]})\|_2 + \|\mathsf{C}(\theta_*^{[\tau]}) - \theta_{-j}^{[\tau]}\|_2.$$

The first term is at most $L_{\text{comp}} \|\widehat{\theta}^{[\tau]} - \theta_*^{[\tau]}\|_2$ by Eq. (42). The second term is at most $\Delta_{\text{comp}}$ by Eq. (41). Applying Theorem D.8 proves the result. $\square$

## D.8. Recalibrated Certified Release

For certification, the noise scale must be calibrated to a public upper bound. Let $M_{\max}$ be a public upper bound on the number of occurrences of the target sample before the unlearning request. In deterministic multi-epoch training, $M_{\max}$ is the number of completed passes in which the sample has appeared. Define

$$U_{\max} := M_{\max} G_{\max}.$$

A public multi-epoch batch-level sensitivity bound is

$$\bar{\Delta}_{\text{ME}}(K, M_{\max}) := \frac{\eta_{\max}}{1 - \gamma} \left( \lambda_{\text{reg}} U_{\max} + \frac{\rho}{2} U_{\max}^2 \right) + 2M_{\max} \gamma^{\max\{0, K - M_{\max} + 1\}} G_{\max}. \tag{69}$$

Under the large-gap condition in Corollary D.9, the truncation term can be replaced by the sharper public bound $2M_{\max}\gamma^K G_{\max}$.

Under deferred compensation, the corresponding sample-level public sensitivity bound is

$$\bar{\Delta}^{(\text{multi})}_{\text{sample}}(K, M_{\max}) := \bar{L}_{\text{comp}}\bar{\Delta}_{\text{ME}}(K, M_{\max}) + \bar{\Delta}_{\text{comp}}. \tag{70}$$

**Corollary D.11** (Multi-epoch certified release under deferred compensation). *Assume the conditions of Theorem D.8, Assumption D.2, and the subspace concentration condition*

$$\|(I - \Pi_k)e_{\text{ME}}\|_2 \leq \beta\|e_{\text{ME}}\|_2, \qquad e_{\text{ME}} := \widehat{\theta}^{[\tau]}_{\text{comp}} - \theta^{[\tau]}_{-j}, \tag{71}$$

*where $\Pi_k$ and $\beta$ are public. Release*

$$\widetilde{\theta}^{[\tau]} := \widehat{\theta}^{[\tau]}_{\text{comp}} + \xi, \qquad \xi \sim \mathcal{N}\left(0, (\sigma^{(\text{multi})}_{\|})^2\Pi_k + (\sigma^{(\text{multi})}_{\perp})^2(I - \Pi_k)\right), \tag{72}$$

*with*

$$\sigma^{(\text{multi})}_{\|} = \frac{2\bar{\Delta}^{(\text{multi})}_{\text{sample}}(K, M_{\max})}{\varepsilon}\sqrt{2\log\left(\frac{2.5}{\delta}\right)}, \tag{73}$$

$$\sigma^{(\text{multi})}_{\perp} = \beta\sigma^{(\text{multi})}_{\|}. \tag{74}$$

*Then $\widetilde{\theta}^{[\tau]}$ is $(\varepsilon, \delta)$-indistinguishable from the Gaussian release of the ideal sample-deletion trajectory*

$$\mathcal{M}(\theta^{[\tau]}_{-j}).$$

*In the ideal compensation regime, this yields the same certified sample-level unlearning guarantee as the single-pass analysis, with the single-pass sensitivity replaced by the multi-epoch sensitivity in Eq. (70).*

*Proof.* By Corollary D.10 and the public calibration in Eq. (70),

$$\|\widehat{\theta}^{[\tau]}_{\text{comp}} - \theta^{[\tau]}_{-j}\|_2 \leq \bar{\Delta}^{(\text{multi})}_{\text{sample}}(K, M_{\max}).$$

The remaining argument is identical to the proof of the single-pass certified release theorem. Decompose the deterministic mean shift $e_{\text{ME}}$ into the public subspace $\Pi_k$ and its orthogonal complement. Apply the Gaussian mechanism to the two orthogonal components with privacy budgets $(\varepsilon/2, \delta/2)$ each, using the sensitivity bound in Eq. (70) and the subspace concentration condition in Eq. (71). Basic composition gives the stated $(\varepsilon, \delta)$ guarantee. $\square$

## D.9. Extension to Forget Sets

The above derivation is stated for a single target sample. For a deletion set $U$, define the affected-step set

$$\mathcal{T}_U := \{t : S_t \cap U \neq \emptyset\}.$$

Each affected mini-batch update is skipped at most once, even if the mini-batch contains multiple target samples. Let $M_U := |\mathcal{T}_U|$.

**Corollary D.12** (Forget-set multi-epoch batch-level extension). *The batch-level approximation results in this appendix hold for a forget set $U$ by replacing $\mathcal{T}_j$ with $\mathcal{T}_U$ and $M$ with $M_U$, provided each affected mini-batch update is counted once.*

For sample-level deletion, let $\theta^{[\tau]}_{-U}$ denote the ideal trajectory that excludes all samples in $U$ while retaining benign samples in the affected mini-batches. Define a set-level deferred compensation operator $\mathsf{C}_U$ that reintroduces the benign samples

$$\{S_t \setminus U : t \in \mathcal{T}_U\}.$$

Assume $\mathsf{C}_U$ satisfies public bounds

$$\|\mathsf{C}_U(\theta) - \mathsf{C}_U(\theta')\|_2 \leq \bar{L}_U\|\theta - \theta'\|_2, \qquad \|\mathsf{C}_U(\theta^{[\tau]}_*) - \theta^{[\tau]}_{-U}\|_2 \leq \bar{\Delta}_U.$$

Then the set-level sample sensitivity can be bounded as

$$\bar{\Delta}_{\text{sample},U}^{(\text{multi})}(K, M_U^{\max}) := \bar{L}_U \bar{\Delta}_{\text{ME}}(K, M_U^{\max}) + \bar{\Delta}_U,$$

where $M_U^{\max}$ is a public upper bound on the number of affected mini-batch updates before the request. Calibrating the Gaussian release noise to $\bar{\Delta}_{\text{sample},U}^{(\text{multi})}$ yields certified indistinguishability from the noisy set-deletion reference $\mathcal{M}(\theta_{-U}^{[\tau]})$ under the same Gaussian mechanism proof.

# E. Additional Experimental Results

**Utility-Forget Trade-off Trajectories**  Figure 4 visualizes the utility-forget trade-off trajectories across training epochs. Each point represents performance at a specific epoch, with trajectories showing how methods evolve during training.

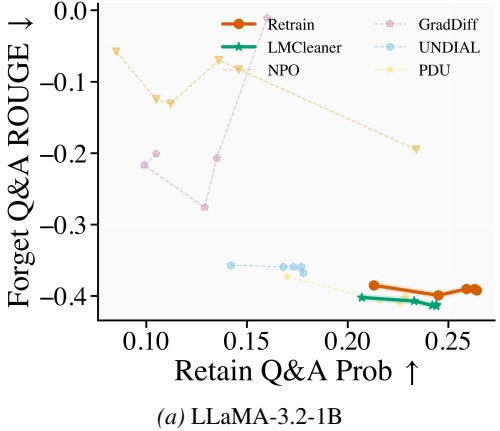 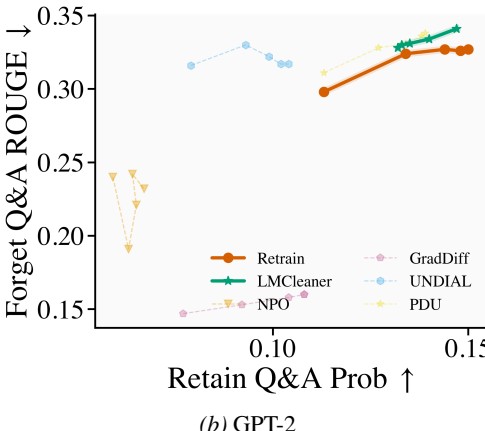

*(a)* LLaMA-3.2-1B  *(b)* GPT-2

*Figure 4.* **Utility-forget trade-off trajectories.** LMCLEANER (green) maintains a compact trajectory closely aligned with Retrain (orange), indicating stable performance across epochs. In contrast, NPO (yellow) exhibits erratic behavior with large trajectory deviations.

LMCLEANER demonstrates remarkable stability: its trajectory remains tightly clustered near Retrain across both models, indicating consistent performance regardless of when the unlearning request arrives. NPO and GradDiff show erratic trajectories, reflecting their sensitivity to the training stage and hyperparameter choices.

**Comprehensive Multi-Metric Comparison**  Figure 5 provides radar chart visualizations comparing all methods across multiple metrics simultaneously. For utility metrics (Model Utility, Retain ROUGE, Retain Prob, Retain Truth, RA ROUGE, WF ROUGE), larger areas indicate better performance. For forget metrics (Forget ROUGE, Forget Prob, Extraction), the interpretation depends on the specific metric. On both models, LMCLEANER shows a more balanced profile than baselines,

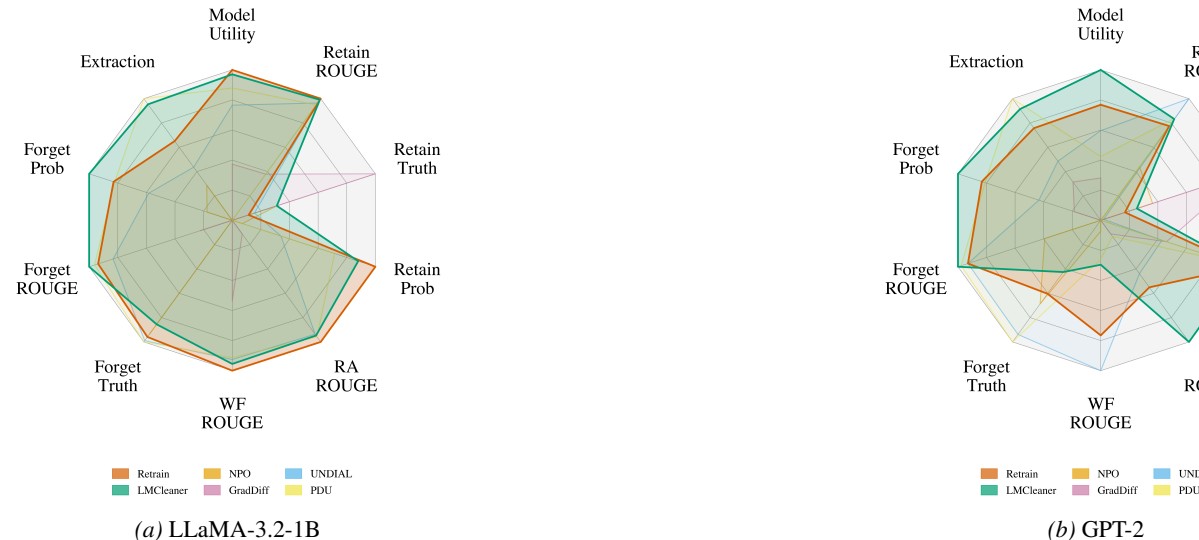

*(a)* LLaMA-3.2-1B  *(b)* GPT-2

*Figure 5.* **Radar chart comparison at Epoch 5.** LMCLEANER (green) achieves comprehensive coverage comparable to Retrain (orange) across all metrics, while baseline methods show significant deficiencies in various dimensions.

with utility metrics close to Retrain. GradDiff (pink) shows collapsed coverage due to utility degradation, while other baselines exhibit uneven profiles with notable weaknesses.

# F. Limitations and Future Work

Several directions remain for future exploration. First, while mini-batch unlearning reduces storage from $O(Np)$ to $O(Tp)$, the trajectory logging overhead remains substantial; sparse checkpointing and gradient compression could further reduce this cost. Second, LMCLEANER's computational advantage is inherently tied to the structure of the forget set: it scales with the number of training batches containing at least one forget sample. In the worst case, an adversary who distributes forget samples such that every training batch is affected can force LMCLEANER to degrade to full retraining cost. Practitioners should be aware that LMCLEANER's efficiency gains are not guaranteed under all forget set distributions. We view this as a design tradeoff: batch-level granularity enables substantial speedups in typical settings, at the cost of vulnerability to adversarial forget set configurations. Third, our theoretical guarantees rely on local strong convexity assumptions that may not hold uniformly across the loss landscape; extending the analysis to handle saddle points and flat regions would strengthen the framework. Fourth, validating LMCLEANER on larger models (7B+ parameters) and diverse benchmarks would further establish its practical applicability. Despite these limitations, LMCLEANER represents a significant step toward practical, certified unlearning for LLMs, enabling real-time compliance with data protection regulations without sacrificing model utility or requiring access to retained training data.

