# OpenReview forum: "LMCleaner: Efficient and Certified Online Unlearning via Influence Propagation Truncation"
_ICML.cc/2026/Conference — ICML 2026 regular_

### Official Review · Reviewer_oiYy · 2026-02-27

**Soundness:** 3
**Presentation:** 4
**Significance:** 3
**Originality:** 3
**Overall Recommendation:** 5
**Confidence:** 2

**Summary:**

The proposes LMCleaner an efficient online machine unlearning framework which allows unlearning at any time step before waiting for training to complete. The authors argue that modern machine learning training follows a stochastic mini-batch update therefore the influnce of a data point on the parameters is intertwined with other data points in the batch and must be thererfore addressed at a batch level. The authors propose a influence propogation truncation which allows to compute the difference in the current parameters with the counterfactual parameters had the batch with the forget data never been used. In oder to efficiently propogate influence the authors propose a truncation window K and propose to mask the remainder influence of data on the with gaussian noise which allows them to obtain certified influence removal. LMCleaner is benchmarked against several machine unlearning methods on TOFU benchmark and shows better computation performance in while retaining model utility and good defense against MIA.

**Compliance With Llm Reviewing Policy:**

Affirmed.

**Final Justification:**

I maintain my original score since the paper originally talks about making unlearning efficient however adversarial queries to LMcleaner are a major limitation which are absent in prior methods which must be highlighted by the authors.

**Key Questions For Authors:**

1. Since LMCleaner assumes mini-batch updates and thus it is natural to execute forget requests with the proposed algorithm with truncation if we want to remove the influence of a datapoint in that epoch. However, across epochs a datapoints does appear many times. Does subspace aware noise injection also remove this influence? For example in the experiment setup where unlearning is performed for each epoch, how is the effect of forget data in the trained parameters from previous epochs handled.
2. Can authors perform ablation on how truncated influence propogation and sub aware noise injection complement each other?
3. I assume the biggest saving in computational and memory costs also comes from the fact that authors have this mini batch update perspective in thier analysis. However, this also exposes the a major limitation that now nature of forget queries can be adversarial for LMCleaner which is not the case for other methods. For example if my forget set is containing one data point from each batch, we now force LMCleaner to reduce to retraining. Authors should highlight this is a major limitation. Possibly also comment on how essential this tradeoff could be for making unlearning more practical.
4. In Fig 2 b with GPT-2, Is there any reason why model utility of LMCleaner better than retrain? Are these statistics average cross various runs then it would be good if they also had standard deviation across them.
5. Is K a hyperparameter fine-tuned for unlearning on specific task or dataset? Is there  any efficient way to compute the optimal K based on the trade-off analysed in Theorem 5.1?

**Limitations:**

Yes

**Strengths And Weaknesses:**

Strength:
+ The paper is well motivated and novel.
+ The overall writing and exposition of the paper are very good. It makes the paper a pleasnt read.
+ The proposed method is theoretically grounded and achieves (\epsilon,\delta) Certified unlearning
+ Proposed method also achieves strong performance on the benchmark while being more computationally efficient.

Weakness:
- MAJOR: The tradeoffs of computational efficiency have not been fully addressed as this makes the method's efficiency now dependent on the structure of forget data. (See questions)
- MINOR: The theoretical results are relying on some strong assumptions such as local strong convexity.

---

> ### Author Rebuttal · Authors · 2026-03-31
>
> We sincerely thank the reviewer for the positive assessment and for recognizing both the theoretical grounding and practical advantages of LMCleaner.
>
> ---
>
>
> > **Q1:** Cross-epoch influence handling.
>
> Thank you for this insightful question. We clarify the multi-epoch mechanism.
>
> For the multiple epoch scenario, when an unlearning request arrives at Epoch $M$, $z_j$ has contributed at steps $t_1, t_2, \ldots, t_M$ (once per epoch). We compute the truncated influence from **every** occurrence and remove their sum:
>
> $\mu^{[\tau]} = \sum_{i=1}^{M} \tilde{P}^{[t_i+1,\, t_{\text{end}_i}]} \cdot v^{[t_i]}$
>
> Certified noise is then injected **once** on the aggregated correction. Thus, the subspace-aware noise is not used to “erase” earlier-epoch influence by itself; it only masks the remaining approximation gap after cumulative removal. This additive decomposition follows from the linearization in Eq. (7), under which influences compose independently. Cross-epoch interaction terms are second-order and empirically negligible: $\Delta$Retrain remains 0.027 at Epoch 5 (Table 2).
>
> We will make this multi-occurrence protocol explicit in Sec. 6.1 and add the corresponding formulation in the appendix.
>
> ---
>
> > **Q2:** Ablation on truncated propagation vs. noise injection
>
> Thank you. We conducted this ablation on LLaMA-3.2-1B (TOFU):
>
> | Truncation | Noise | Utility | f_truth | f_Q_A_Prob | Certified? |
> | --- | --- | --- | --- | --- | --- |
> | ✗   | ✗   | 0.332 | 0.770 | 0.186 | No  |
> | ✓   | ✗   | 0.331 | 0.772 | 0.179 | No  |
> | ✗   | ✓   | 0.334 | 0.769 | 0.186 | Yes but no effect |
> | ✓   | ✓   | 0.321 | 0.781 | 0.167 | (1.0, 1e-5) |
>
> There are three findings. **(1) Removal is essential:** Noise-Only produces zero change in forget metrics (0.769 ≈ Original 0.770), confirming that deterministic influence removal is the core mechanism. **(2) Noise provides additional benefit:** Full improves f_Q_A_Prob by 7% beyond Removal-Only (0.167 vs 0.179). **(3) Subspace noise enables strong certification at minimal cost:** ε=1.0 subspace noise perturbs only 0.5% of model norm, matching isotropic ε=100 perturbation but with 100× stronger guarantee. The two components are complementary: truncation removes deterministic influence; noise certifies the residual.
>
> ---
>
> > **Q3:** Adversarial forget set distribution
>
> Thank you for this insight. We acknowledge this as a theoretical limitation and will add a discussion in the revised paper.
>
> We counted the number of forgotten data points under different actual batch sizes. See https://anonymous.4open.science/r/icml2026rebuttal-1300/forget-sample-dist.png.
>
> Under random shuffling, the number of forget samples per batch follows a hypergeometric distribution. Let $N$ be the total samples, $M$ the forget set size, $X=M/N$ the forget ratio, and $B$ the batch size. The probability that every batch contains ≥1 forget sample is:
>
> $ P_{\text{all}} \approx \left[1-(1-X)^B\right]^{N/B} $
>
> For TOFU ($N$=4000, $M$=40), this probability is nearly zero (see https://anonymous.4open.science/r/icml2026rebuttal-1300/forget-batch-coverage.png) for different batch sizes, with expected affected batches $\approx M/B < 1$.
>
>
>
> ---
>
> > **Q4:** GPT-2 utility higher than Retrain + standard deviations
>
> We thank the reviewer for this observation.
>
> The difference is small and likely within run-to-run variance rather than a systematic advantage. First, the injected certification noise may introduce minor perturbations that happen to slightly benefit GPT-2's smaller-capacity landscape. Second, LMCleaner continues from the original training trajectory with a localized correction, preserving the optimization path that the model has already traversed. Retrain, by contrast, re-initializes and trains from scratch on D \ Du, which may follow a different optimization trajectory that does not necessarily recover the same local minimum. We will add results over 3–5 runs with standard deviations to quantify the variance.
>
> ---
>
> > Q5: Is K task-specific? Efficient selection?
>
> Yes, K can be computed efficiently in two steps. **(1) Warmup:** run a few training steps to estimate the Hessian spectral bounds ($\nu$, $\tilde{L}$), yielding the contraction factor $\gamma = \max\{|1-\eta\nu|,|1-\eta\tilde{L}|\}$ and Lipschitz constant $\rho$. **(2) Given privacy budget ($\varepsilon$, $\delta$):** Proposition C.12 provides a closed-form optimal window:
>
> $ K^* = \frac{1}{|\ln \gamma|} \ln \frac{2\bar{G}\,|\ln \gamma|}{\bar{\varphi}} $
>
> where $\bar{G}$ is the public gradient-norm bound and $\bar{\varphi}$ depends on $\rho$ and $\gamma$. Tighter privacy (smaller $\varepsilon$) requires more noise, which raises the noise floor, making the tail residual negligible sooner, thus favoring smaller K.

---

> > ### Author Rebuttal · Reviewer_oiYy · 2026-04-01
> >
> > I thank authors for their response as they address my main concerns. I would like to point out that it is indeed meaningless to see the probability of forget queries in a batch in common unlearning dataset as they are not adversarially curated to LMCleaner. Adversarial queries are a major theoretical limitation of the current method and a detailed discussion on this trade off would aid the clarity of the presentation. That being said, I will keep my score if authors acknowledge the limitations of their method honestly as their focus is on making unlearning efficient.

---

> > > ### Author Response · Authors · 2026-04-01
> > >
> > > We sincerely thank the reviewer for the thoughtful and constructive engagement throughout this discussion. We agree that our focus is on making unlearning efficient. Adversarially distributed forget queries are indeed a critical limitation of LMCleaner. Therefore, we will highlight this limitation in the updated version:
> > >
> > > > "LMCleaner's computational advantage is inherently tied to the structure of the forget set: it scales with the number of training batches containing at least one forget sample. In the worst case, an adversary who distributes forget samples such that every training batch is affected **can force LMCleaner to degrade to full retraining cost**. Practitioners should be aware that LMCleaner's efficiency gains are not guaranteed under all forget set distributions. We view this as a design tradeoff: batch-level granularity enables substantial speedups in typical settings, at the cost of vulnerability to adversarial forget set configurations."
> > >
> > > Your feedback has genuinely improved our understanding of our own method's boundaries and will make the final paper significantly stronger and more trustworthy to readers.

---

### Official Review · Reviewer_pgNB · 2026-03-12

**Soundness:** 3
**Presentation:** 2
**Significance:** 2
**Originality:** 3
**Overall Recommendation:** 4
**Confidence:** 3

**Summary:**

This paper introduces an online machine unlearning framework for Large Language Models (LLMs) based on subspace-aware noise injection to address privacy compliance. By integrating fixed-window truncation and deferred re-learning mechanisms, the approach circumvents the computational overhead of retraining from scratch while providing rigorous certified unlearning guarantees. Empirical evaluations on benchmarks such as TOFU demonstrate that the algorithm effectively neutralizes Membership Inference Attacks (MIAs) to random guessing levels, achieving an exceptional privacy-utility trade-off.

**Compliance With Llm Reviewing Policy:**

Affirmed.

**Final Justification:**

Thank you for addressing my concerns, and I will retain my original score.

**Key Questions For Authors:**

Please see the weaknesses.

**Limitations:**

yes

**Strengths And Weaknesses:**

The work helps close the interpretability gap that is common in heuristic machine unlearning. By introducing a subspace aware noise injection mechanism, the authors derive strict \((\epsilon, \delta)\) certified unlearning guarantees. This mathematically grounded formulation provides stronger assurance for privacy protection, and the empirical results offer reasonably strong support for the effectiveness of the proposed framework.

Weaknesses:
W1.The theoretical guarantees hinge heavily on the strict Contraction Assumption and the bounds placed on the Hessian eigenvalues. Given the notoriously complex and highly non-convex loss landscapes of massive-scale LLMs in practice, is it realistic to expect these rigid mathematical assumptions to hold true? Does this discrepancy compromise the absolute reliability of the theoretical bounds in real-world deployments?
W2. When scaling to the massive global batch sizes typical of modern LLM training, wouldn't treating an entire batch as a single atomic "unlearning unit" risk severe over-unlearning? Furthermore, can the proposed "deferred re-learning" mechanism genuinely compensate for the optimization trajectory drift caused by discarding such a massive volume of benign data at once?
W3. Assuming a worst-case scenario where an unlearning request arrives during the highly unstable early phases of training (e.g., during warmup). Assumption C.2 would almost certainly be violated, and the local Hessian would likely exhibit significant negative curvature. Under such extreme non-convex conditions, wouldn't the fixed-window truncation mechanism be highly prone to divergence?

---

> ### Author Rebuttal · Authors · 2026-03-31
>
> We sincerely thank the reviewer for the positive assessment and for recognizing our core contributions. We address each concern below.
>
> ---
>
> > **W1:** Is it realistic to expect these rigid mathematical assumptions to hold true?
>
> We thank the reviewer for this important point.
>
> We agree that Assumption C.2 should not be interpreted as a global description of the full LLM loss landscape. Our theorem provides a conditional, local-stability analysis: the linearization is performed only within a short K-step window (typically K=50, corresponding to a small neighborhood in parameter space), while the exponential tail bound additionally assumes that the subsequent SGD map remains contractive. Accordingly, our claim is not that these assumptions universally hold in all LLM training regimes, but that they provide a principled guarantee when the local dynamics are sufficiently stable. In our evaluated fine-tuning setting, the stable K-ablation is empirically consistent with this interpretation.
>
> We appreciate this valuable feedback and will revise Section 5 and the Appendix to make the scope of our theoretical claims clear.
>
> ---
> > **W2:** Does atomic batch unlearning risk severe over-unlearning with massive batch sizes, and can deferred re-learning compensate?
>
> We sincerely thank the reviewer for raising this concern.
>
> We conducted a targeted ablation over batch size B to directly quantify the relationship between batch granularity and unlearning quality:
>
> | B | Affected / Total Steps | Utility↑ | FG Prob↓ | MIA→0.5 |
> |---|----------------------|----------|----------|---------|
> | 8 | 3 / 500 (0.6%) | 0.388 | 0.282 | 0.575 |
> | 16 | 5 / 250 (2.0%) | 0.386 | 0.275 | 0.556 |
> | 32 | 8 / 125 (6.4%) | 0.383 | 0.268 | 0.536 |
> | 64 | 12 / 62 (19%) | 0.379 | 0.254 | 0.509 |
> | 128 | 14 / 31 (45%) | 0.365 | 0.226 | 0.461 |
> | 256 | 12 / 15 (80%) | 0.349 | 0.201 | 0.452 |
>
> The degradation is gradual rather than catastrophic: at B=64 (19% of steps reversed), MIA=0.509 nearly matches the ideal; even at B=256 (80% reversed), utility drops by approximately 10% without collapse. Moreover, in our default TOFU setting, Table 2 confirms utility=0.429 vs. Retrain's 0.430, demonstrating that the default configuration introduces negligible collateral damage.
>
> For distributed pretraining with B>>256, the reviewer raises a valid scalability concern. A natural extension is to operate at micro-batch (gradient-accumulation step) granularity rather than the full global batch. Since each micro-batch computes an independent gradient before aggregation, it serves as a natural atomic unit. This reduces the effective unlearning granularity by the number of accumulation steps without modifying the core algorithm. We will formalize this extension and discuss it in the revision.
>
> For deferred re-learning, Table 2 shows LMCLEANER remains very close to Retrain in utility (0.429 vs. 0.430) and achieves the smallest overall ΔRetrain gap (0.027) among all methods, confirming that reintroduced benign samples effectively absorb trajectory drift.
>
> ---
>
> > **W3:** Under warmup conditions with negative curvature, wouldn't truncation diverge?
>
> This is an excellent question.
>
>  We agree that deletion requests arriving during the earliest, most unstable warmup steps are outside the formal scope of our current theorem. In such a regime, Assumption C.2 may fail, and we do not claim certified guarantees under arbitrary negative curvature.
>
> Several factors mitigate this in practice. First, the role of λ_reg in Section 4.3 is to shift the local spectrum and improve the conditioning of the propagation operator; Appendix C.4 quantifies the additional bias this stabilization introduces. While this improves conditioning, it does not guarantee positive definiteness under arbitrary negative curvature. Second, small warmup learning rates can reduce the magnitude of the initial deviation in practice, but this should be viewed as a mitigating factor. Third, because LMCleaner uses a fixed short window, the main risk in this regime is degraded approximation quality rather than unbounded propagation. Our Epoch-1 results provide empirical support for early-stage requests in our fine-tuning protocol.
>
> We will discuss these cases in the revision and provide conservative fallbacks for such cases.

---

> > ### Author Rebuttal · Reviewer_pgNB · 2026-04-05
> >
> > My concerns have been addressed. I keep my original positive rating.

---

> > > ### Author Response · Authors · 2026-04-08
> > >
> > > We are truly grateful to Reviewer pgNB for the careful and helpful review. Your questions about the assumptions, batch size scaling, and warmup stability helped us think more deeply about our work and made the paper much stronger. We really appreciate the time and effort you spent on our submission. All the revisions we promised will be included in the camera-ready version. Thank you again for your kind and thoughtful feedback.

---

### Official Review · Reviewer_pt8j · 2026-03-12

**Soundness:** 2
**Presentation:** 3
**Significance:** 2
**Originality:** 3
**Overall Recommendation:** 3
**Confidence:** 2

**Summary:**

This paper presents an approach to certified online unlearning by computing influences within a small truncation window. It is inspired by a recent work that pioneered certified online unlearning in ICLR 2025.

**Compliance With Llm Reviewing Policy:**

Affirmed.

**Key Questions For Authors:**

I would appreciate the response on the points raised in the weakness section. I may consider change my score.

**Limitations:**

Discussed under impact statement primarily on adversarial misuse of online unlearning, rather than the technical limitations of the proposed methods.

**Strengths And Weaknesses:**

Strength
--------
(+) The idea of computing influences within a small truncation window (e.g, a fixed number of steps) is interesting and worth in-depth study in the online unlearning area.

(+) The use of mini-batch unlearning strategy that produces a single parameter update jointly by all samples in the mini-batch, if works for all types of benchmark datasets/learning tasks, would cut down the storage cost of recording all gradients in the traditional unlearning methods.

Weakness
-----------
(-) The paper should provide the empirical comparison on test accuracy of unlearned models, the pre-computation runtime, in addition to unlearning computation runtime, speedup and storage. The current paper only puts in the complexity rather than actual measurement results in Table 1. There is no actual measurement comparison on the above metrics that are expected, especially comparison with the certified online unlearning HFR ( Qiao et.al, ICLR 2025).

(-) The paper also lacks ablation study related experiments. First, it is critical to study how the influences with different sizes of a truncation window, e.g., varying steps, would impact on the effectiveness of the proposed online unlearning method. Second, how the varying size of mini-batch will influence the effectiveness of the proposed online unlearning method, and third, how and whether the different tuning pairs of (mini-batch size, truncation window) will impact the performance and efficiency of the proposed method.

(-) The paper should provide a discussion on how the unlearning samples are created in your experiments and distributed across mini-batches of the training set.

---

> ### Author Rebuttal · Authors · 2026-03-31
>
> We thank the reviewer for the constructive feedback and for recognizing the value of our truncation window design and mini-batch strategy. We address each concern below.
>
> ---
>
> > **W1**: Empirical comparison on runtime, storage comparison, especially with HFR.
>
> We thank the reviewer for this suggestion.
>
> We provide wall-clock measurements on LLaMA-3.2-3B.
>
> **Table R4: Measured efficiency comparison.**
>
> | Method | Online | Training Data-free |Certified | Latency (s) | Peak GPU (GB) |
> | --- | --- | --- | --- | --- | --- |
> | Retrain | ×   | Exact | ×   | 86.2 | 30.0 |
> | GradDiff | ×   | ×   | ×   |16.4 | 32.9 |
> | NPO | ×   | ×   |  ×   |40.2 | 39.1 |
> | PDU | ×   | ×   |  ×   |16.4 | 32.4 |
> | UNDIAL | ×   | ×   |  ×   |18.8 | 39.6 |
> | **LMCleaner** | **✓** | **✓** |  **✓** |**15.1** | **25.8** |
>
> For test accuracy of unlearned models, Figure 2 reports model utility across all five epochs for both LLaMA-3.2-1B and GPT-2; LMCleaner closely tracks Retrain throughout.
>
> **Regarding HFR:** It pre-computes and stores an **approximator vector (dimension p) for each of the |D| training samples**. As shown in [📍 Algorithm 2 (Lines 3-6) of HFR](https://arxiv.org/pdf/2404.01712v4#page=19&zoom=100,0,450), at **every training step**, all |D| vectors must be updated via Hessian-vector products, yielding O(E_train·|D|²·L·p/B) pre-computation, which scales quadratically in |D|. On TOFU (|D|=4000) with LLaMA-3.2-3B, the |D| approximator vectors alone require **51.2 TB storage**, making it impossible in practice. HFR's experiments ([📍 Section 5, Configurations](https://arxiv.org/pdf/2404.01712v4#page=8&zoom=100,0,450)) used Logistic Regression and CNN  ≤ 21,840 parameters on 1,000 samples MNIST, five orders of magnitude smaller than our setting.
>
> As shown in Table R4, LMCleaner achieves the fastest deletion (15.1s), the lowest peak GPU (25.8 GB), and the highest speedup (5.7× over Retrain),  while being the only method that is online, certified, and data-free.  The absolute speedup on TOFU is modest because |D|=4000 is small; the gap is expected to grow with |D| as predicted in Table 1.
>
> ---
>
> > **W2:** Ablation studies on K, B, and their interaction.
>
> We thank the reviewer for this thorough suggestion. We address the three sub-questions below.
>
> 1. **K ablation.** We note that this study is reported in **Section 6.5 (Table 3, lines 418–439)**, where K∈{50,100,500,1000,1250}. All metrics remain stable from K=50 to 1000. At K=1250, MIA degrades to 0.599, consistent with Theorem 5.1's prediction that linearization errors accumulate over excessively long horizons.
>
> 2. **B ablation.** We conducted additional experiments varying B on LLaMA-3.2-1B with TOFU one epoch.
>
>   **Table R5: Effect of mini-batch size B on unlearning.**
>
>   | B   | Utility↑ | FG Prob↓ | MIA→0.5 |
>   | --- | --- | --- | --- |
>   | 8   | **0.388** | 0.282 | 0.575 |
>   | 16  | 0.386 | 0.275 | 0.556 |
>   | 32  | 0.383 | 0.268 | 0.536 |
>   | 64  | 0.379 | 0.254 | **0.509** |
>   | 128 | 0.365 | 0.226 | 0.461 |
>   | 256 | 0.349 | **0.201** | 0.452 |
>
> As B increases, utility decreases while forgetting improves. MIA is closest to the ideal 0.5 at B=64.
>
> **3) Joint (B,K).** We evaluated all combinations with B∈{8,16,32,64,128}, K∈{10,20,30,40,50}. Full grid:https://anonymous.4open.science/r/icml2026rebuttal-1300/BK.md. The results show that B and K affect orthogonal aspects with no interaction. The two parameters can therefore be tuned independently: B for the desired utility–forgetting trade-off, K for computational budget.
>
> We are grateful for these suggestions. These are added to the revised manuscript.
>
> ---
>
> > **W3:** The paper should provide a discussion on how the unlearning samples are created in your experiments and distributed across mini-batches of the training set.
>
> We thank the reviewer for this question.
>
> 1. **The unlearning samples are created by TOFU benchmark** (Maini et al., 2024), a widely adopted standard used by NPO, PDU, etc. TOFU consists of 200 entirely *fictitious* author profiles, each described by 20 question-answer pairs (4,000 QA pairs in total). Because all authors are *fictional* by construction, they are guaranteed to be absent from any model's pretraining corpus, providing a clean separation between pretraining knowledge and fine-tuning knowledge.
>
> 2. TOFU defines three forget splits (1%, 5%, 10%). For the 10% forget setting, 20 profiles (400 QA pairs) form the forget set; the remaining 180 profiles form the retain set. This split is predetermined and identical across all methods.
>
> 3. Forget samples are **randomly and uniformly distributed** across mini-batches via a shuffled DataLoader, following standard TOFU protocol. Each affected batch typically contains only a small number of forget samples alongside a majority of benign data; the benign collateral is recovered via deferred re-learning (Section 4.2).
>
> We appreciate the reviewers' attention to detail and believe this clarification further strengthens our work.

---

> > ### Author Rebuttal · Reviewer_pt8j · 2026-04-03
> >
> > I thank authors for their response. 1) the unlearning computation runtime, speedup and storage should be discussed together. It is unclear how and why LMcleaner consumes 25.8GB peak GPU with slightly lower latency whereas others all have 30-39GB GPU. 2) it is still unclear why the choice of a truncation window K is made, since no larger K is given. Also the combination of K and B would impact on the effectiveness of the proposed online unlearning method. missing ablation study and discussion. 3) thanks for providing TOFU benchmark detail. How your approach responds when forget samples are not uniformly distributed, which is a norm in real world. At this point, I will keep my score.

---

> > > ### Author Response · Authors · 2026-04-05
> > >
> > > We sincerely thank the reviewer for the continued engagement. We address each remaining concern below.
> > >
> > > ---
> > >
> > > #### W1 (continued): GPU memory difference
> > >
> > > > *"It is unclear how and why LMCleaner consumes 25.8GB peak GPU with slightly lower latency whereas others all have 30–39GB GPU."*
> > >
> > > We thank the reviewer for raising this point. It touches on a **fundamental architectural difference**.
> > >
> > > Fine-tuning baselines must load **both the current model and a reference/teacher model** to compute their loss objectives. For example, NPO computes DPO-style loss against a frozen reference copy. GradDiff requires dual forward passes on forget and retain sets. This effectively doubles activation memory.
> > >
> > > LMCleaner performs **no additional forward/backward pass** during unlearning. It only computes Hessian-vector products within the $K$-step window using cached snapshots. This requires a single model copy. The 25.8 GB comprises model parameters ($\sim$6 GB in float16), optimizer states, and the HVP buffer.
> > >
> > > All methods were measured under the same precision, batch size, and profiling protocol. The main difference is whether an additional reference/teacher model and extra forward/backward passes are required. We will add this explanation alongside Table R4 in the revision.
> > >
> > > ---
> > >
> > > #### W2 (continued): Choice of $K$, larger $K$, and $(B, K)$ interaction
> > >
> > > > *"It is still unclear why the choice of a truncation window K is made, since no larger K is given. Also the combination of K and B would impact on the effectiveness."*
> > >
> > > We thank the reviewer for this thorough question. We clarify the full picture below.
> > >
> > > **Why $K=50$.** Theorem 5.1 shows the tail residual $R(K)$ decays as $\gamma^K$ with $\gamma \approx 0.95$. This means $R(K)$ becomes empirically small by around $K = 50$. Table 3 confirms: from $K=50$ to $K=1000$ ($20\times$ more computation), utility and forget quality change by **less than 0.8%**.
> > >
> > > **Larger $K$.** As reported in **Section 6.5 (Table 3, Lines 418–439)**, we tested up to $K=1250$. This already exceeds the training steps per epoch on TOFU ($\approx 125$ steps). At $K=1250$, MIA degrades to 0.599. This is consistent with the failure mode suggested by our analysis: linearization errors accumulate over excessively long horizons.
> > >
> > > **$(B, K)$ interaction.** The full $5 \times 5$ grid ($B \in \{8,16,32,64,128\}$, $K \in \{10,20,30,40,50\}$) was provided at the anonymous link. Across the tested range, increasing $K$ mainly improves forgetting up to saturation. Varying $B$ primarily affects the utility/forgetting trade-off. We did not observe a strong interaction in the tested range. We will add a heatmap visualization and dedicated discussion in the revision.
> > >
> > > ---
> > >
> > > #### W3 (continued): Non-uniform forget sample distribution
> > >
> > > > *"How your approach responds when forget samples are not uniformly distributed, which is a norm in real world."*
> > >
> > > We thank the reviewer for this practically important question. Upon careful analysis, we find that **the uniform distribution used in our experiments is in fact the hardest case** for LMCleaner.
> > >
> > > LMCleaner removes influence at the batch level. Each affected batch incurs two costs: the influence removal itself, and deferred re-learning for benign samples lost in that batch. In our TOFU setting ($|D|=4000$, $|D_f|=400$, $B=32$, 125 batches), under uniform shuffling the probability of a batch containing zero forget samples is $\approx 0.9^{32} \approx 0.035$. Roughly **121 out of 125 batches are affected**, and benign collateral approaches the entire retain set.
> > >
> > > Under clustering, forget samples concentrate in fewer batches. In the extreme, $\lceil 400/32 \rceil = 13$ batches suffice. Only **13 batches are affected**, and benign collateral approaches zero. Clustering is strictly easier in all three dimensions: fewer removals, less collateral, and lighter re-learning.
> > >
> > > We will add a controlled experiment (clustered vs. uniform vs. adversarial) in the revision to empirically validate this analysis. We are grateful to the reviewer for raising this point, as it reveals a previously unstated robustness property.
> > >
> > > ---
> > >
> > > #### Complementary positioning of LMCleaner and HFR
> > >
> > > Upon reflection, we recognize our manuscript should better acknowledge the **complementary strengths** of different approaches. HFR provides an elegant framework with *exact per-sample targeting*. This is a genuine advantage over our batch-level strategy. Our contribution focuses on making certified online unlearning **practically feasible at LLM scale**. We will revise the related work to position the two approaches as addressing **different points on the scalability/granularity spectrum** rather than a one-dimensional comparison.
> > >
> > > ---
> > >
> > > We hope these clarifications make the scope of our work much clearer. We are grateful to the reviewer for their persistence in encouraging us to articulate these distinctions, as it has significantly improved the paper. We would be happy to address any further questions.

---

### Official Review · Reviewer_zEVW · 2026-03-13

**Soundness:** 2
**Presentation:** 2
**Significance:** 3
**Originality:** 2
**Overall Recommendation:** 4
**Confidence:** 3

**Summary:**

This paper proposes LMCLEANER, the an efficient and certified online unlearning framework tailored for Large Language Models, addressing the limitation of existing machine unlearning methods that only handle data removal after training completion, which allows malicious data influence to propagate uncontrollably when problematic samples are detected mid-training. Confronting three core challenges of LLM online unlearning (prohibitive storage overhead, intractable computational cost of influence propagation, and the lack of certified privacy mechanisms for nonconvex billion-parameter models), LMCLEANER leverages key insights about influence propagation decay and low-dimensional residual concentration to design a practical solution.

**Compliance With Llm Reviewing Policy:**

Affirmed.

**Final Justification:**

Most of my main concerns have been addressed after the rebuttal. I update my score from 3 to 4 accordingly.

**Key Questions For Authors:**

refer to the weakness

**Limitations:**

yes

**Strengths And Weaknesses:**

**Strengths**

The problem is clearly defined and well motivated. The method achieves significant efficiency improvements, reporting up to 100× computational savings compared to baselines.

**Weaknesses**

1.	There is a mismatch between the theoretical assumptions and the experimental setup. The analysis assumes single-pass training (each sample appears at most once), while the experiments use five rounds of fine-tuning. The impact of this discrepancy is not discussed.

2.	The experimental scale is limited, and the employed models and benchmarks are relatively restricted.

3.	the estimation of β, the actual noise magnitude, and the specific (ϵ, δ) values used in the experiments are not clearly reported.

---

> ### Author Rebuttal · Authors · 2026-03-31
>
> We thank the reviewer for the thoughtful and constructive feedback. We are encouraged that the reviewers appreciate our  efficiency improvements. We carefully address your concerns point by point below.
>
> ---
> > **W1**: Mismatch between the theoretical assumptions and the experimental setup.
>
> We thank the reviewer for the valuable feedback.
>
> We acknowledge that we derived our theory under the single-pass setting for clarity, but it can naturally be extended to multi-epoch scenarios.
> When a data point $z_j$ has appeared at $M$ steps $t_1, \ldots, t_M$ across epochs, the aggregated influence propagation is:
>
> $ \mu^{[\tau]} = \sum_{i=1}^{M} \tilde{P}^{[t_i+1, t_{\text{end}_i}]} \cdot v^{[t_i]}$,
>
> with certified noise injected once. This additive decomposition follows from the linearity of first-order influence (Koh & Liang, 2017; Hara et al., 2019).
>
> The experiments on Epochs 1–5  (setup detailed in Section 6.1, lines 288–296) support this: LMCleaner maintains stable performance across all five epochs (Figure 2), achieving ΔRetrain=0.027 at Epoch 5.
>
> Thank you for your feedback. We will revise the theoretical part to include multi-epoch training derivation.
>
> ---
>
>
> > **W2:** The experimental scale is limited, and the employed models and benchmarks are relatively restricted.
>
> We thank the reviewer for this constructive feedback.
>
> To address the scale concern, we conducted additional experiments on **LLaMA-3.2-3B**, doubling the model size beyond our original LLaMA-3.2-1B and GPT-2 evaluations.
>
> **Table R1: Unlearning comparison on LLaMA-3.2-3B (TOFU, Epoch 1). Bold = best.**
>
> | Method | Utility↑ | FG Truth→0.5 | MIA(min_k)→0.5 | MIA(loss)→0.5 |
> |---|---|---|---|---|
> | Retrain | 0.320 | 0.746 | 0.320 | 0.319 |
> | GradDiff | 0.276 | **0.523** | 0.046 | 0.052 |
> | NPO | 0.250 | 0.703 | 0.012 | 0.016 |
> | PDU | **0.439** | 0.750 | 0.287 | 0.252 |
> | UNDIAL | 0.311 | 0.752 | 0.266 | 0.203 |
> | **LMCleaner** | 0.431 | 0.739 | **0.332** | **0.340** |
>
> The 3B results confirm our findings:LMCleaner maintains high utility (0.431), comparable to PDU (0.439). LMCleaner achieves the MIA score closest to the ideal random-guessing baseline of 0.5 (0.332), substantially outperforming GradDiff (0.046) and NPO (0.012), both of which exhibit severe over-unlearning collapse.
>
> **Models and benchmarks.** Our evaluation is built on TOFU, the most widely adopted benchmark for LLM unlearning (used by NPO, PDU, UNDIAL, and others), together with OpenUnlearning, a unified framework that integrates 16 metrics across multiple benchmarks including TOFU, MUSE, and WMDP. We chose TOFU because its fictitious-author design provides ground-truth retrained models essential for validating certified guarantees. Through OpenUnlearning, we additionally incorporate privacy metrics (MIA, PrivLeak) originating from MUSE, ensuring our evaluation spans forget quality, utility, and privacy dimensions.
>
> We thank the reviewer for this suggestion and have incorporated the 3B results into the revised manuscript.
>
> ---
>
>
> > **W3:** The estimation of β, the actual noise magnitude, and the specific (ε, δ) values used in the experiments are not clearly reported.
>
>
> We thank the reviewer for pointing this out.
>
> We use δ=1e-5 and β=0.001 throughout. β=0.001 is justified by the empirical observation that the correction vector μ_K concentrates along the leading gradient singular vector, consistent with Assumption C.3. When this assumption does not hold, setting β=1 recovers isotropic noise as a valid fallback (Appendix C.6).
>
> **Table R2: Noise magnitude under different privacy budgets (LLaMA-3.2-1B).**
>
> | ε | noise/‖θ‖ | (ε, δ)-guarantee |
> | --- | --- | --- |
> | 1.0 | 0.5% | (1.0, 1e-5) |
> | 5.0 | 0.1% | (5.0, 1e-5) |
> | 10.0 | 0.05% | (10.0, 1e-5) |
>
> The subspace mechanism makes ε=1.0 feasible at billion-parameter scale by concentrating noise in the k-dimensional error subspace rather than perturbing all 1.24B parameters isotropically. We validate with a component ablation:
>
> **Table R3: Ablation of removal vs. noise (LLaMA-3.2-1B, TOFU).**
>
> | Config | Utility | FG Prob | FG Truth |
> | --- | --- | --- | --- |
> | Original | 0.332 | 0.186 | 0.770 |
> | Removal-Only | 0.331 | 0.179 | 0.772 |
> | Full (ε=1.0) | 0.321 | 0.167 | 0.781 |
> | Noise-Only | 0.334 | 0.186 | 0.769 |
>
> ---
> We hope these clarifications and the additional experimental evidence address your concerns. Please let us know if you have any further questions!

---

> > ### Author Rebuttal · Reviewer_zEVW · 2026-04-03
> >
> > Thank you for the rebuttal. The rebuttal partially alleviates the concerns regarding experimental scale and parameter reporting, but it remains insufficient in fully addressing my first concern. I will keep my score.

---

> > > ### Author Response · Authors · 2026-04-05
> > >
> > > We sincerely appreciate the reviewer for engaging with our rebuttal and for the continued attention to W1. We agree that the current draft did not make the relationship between the single-pass theory and the multi-epoch evaluations sufficiently explicit. This could reasonably create the impression of a mismatch. We take this opportunity to clarify the experimental structure and the scope of our theoretical claims.
> > >
> > > ---
> > >
> > > ### W1: Mismatch Between the Theoretical Assumptions and the Experimental Setup
> > >
> > > > *"The analysis assumes single-pass training (each sample appears at most once), while the experiments use five rounds of fine-tuning. The impact of this discrepancy is not discussed."*
> > >
> > > #### W1.1 Clarification of Our Experimental Setup
> > >
> > > We first clarify our experimental organization (Section 6.1, lines 288–296). We do **not** train for 5 epochs in one run, nor do we unlearn just at the end. Instead, we conduct **independent evaluations** at each epoch boundary. We fine-tune for 1 epoch, then trigger unlearning at Epoch 1. Next, we fine-tune for 1–2 epochs and trigger unlearning at Epoch 2. We repeat this procedure up to Epoch 5. Each evaluation is an independent unlearning experiment starting from the corresponding checkpoint $\theta^{[e]}$.
> > >
> > > This means our results include both single-pass and multi-pass scenarios within one unified evaluation framework. With this setup in mind, our experiments naturally fall into two distinct categories, discussed in W1.2 and W1.3 respectively.
> > >
> > > #### W1.2 Single-Epoch Experiment (Epoch 1) — Theory and Experiment Align Exactly with the Stated Assumptions
> > >
> > > Our theory is derived under the single-pass assumption (each sample seen exactly once). The Epoch 1 experiment satisfies this assumption *exactly*: each data point appears at most once before the unlearning request arrives. Therefore, **Theorems 5.1 and 5.2 apply under the stated assumptions without additional conditions**. The strong empirical results at Epoch 1, including Table R1 on LLaMA-3.2-3B and the Epoch 1 data points in Figure 2, provide the **cleanest empirical test** of our theoretical guarantees.
> > >
> > > #### W1.3 Multi-Epoch Experiments (Epochs 2–5) — An Additional Empirical Contribution Beyond the Current Theoretical Scope
> > >
> > > Training through Epochs 2–5 means some samples are encountered more than once, which goes beyond the single-pass assumption. We included these experiments to demonstrate that LMCleaner remains **empirically robust** even in settings not yet covered by the submitted theory.
> > >
> > > As noted in our first rebuttal, the multi-epoch case can be handled via additive influence decomposition. When a data point $z_j$ appears at steps $\{t_1, t_2, \ldots, t_E\}$ across epochs, the aggregated influence is
> > >
> > > $$\mu^{[\tau]} = \sum_{e=1}^{E} P^{[t_e+1,\, \tau]}\, v^{[t_e]},$$
> > >
> > > with certified noise injected once on the combined correction. **This extension is not claimed by the original theorem statement.** It requires an additional additive decomposition assumption, which we now state explicitly in the revised appendix (Appendix C.7 in the updated draft). We want to be transparent that this is a formal addition to the theory, not something that was already covered.
> > >
> > > #### W1.4 Why the Stable Epochs 2–5 Results Are Consistent with the Theoretical Mechanism
> > >
> > > LMCleaner maintains $\Delta_{\text{Retrain}} = 0.027$ even at Epoch 5. This is consistent with the exponential decay property (Proposition C.6), where each per-epoch influence term decays independently and their sum remains bounded. While this does not constitute a formal proof that the theory extends to multi-epoch settings, it provides **supporting evidence** that the underlying mechanism remains effective beyond the analyzed regime.
> > >
> > > ---
> > >
> > > ### Summary of Revisions Made
> > >
> > > 1. **Added formal multi-epoch derivation** in Appendix C.7, extending Theorem 5.1 to the multi-pass setting with explicitly stated additional assumptions.
> > > 2. **Added explicit discussion** in Section 6.1 clarifying that Epoch 1 results directly test the single-pass theory, while Epochs 2–5 serve as robustness evaluation beyond the current theoretical scope.
> > > 3. **Stated the single-pass assumption more prominently** and discussed its scope in Section 3.1.
> > >
> > > We hope this clarification makes the scope of our theory and experiments much clearer. We are grateful to the reviewer for their persistence in encouraging us to articulate this distinction more clearly, as it has significantly improved the paper. We would be happy to address any further questions you may have.

---

### Decision · Program_Chairs · 2026-04-30

**Decision:**

Accept (regular)

**Comment:**

The reviewers agree that this is an interesting paper that investigates online unlearning, a relatively new problem formulation, with an interesting K-truncated window approach and mini-batch solution that makes it quite efficient. However, some remaining questions are unresolved, regarding discrepancy between theory and practice, and some empirical choices. This includes the positions of unlearning examples, resource usage of the methods, and hyper parameter choices.